# PhysDiff: A Physically-Guided Diffusion Model for Multivariate Time Series Anomaly Detection

**Long Li**[1,*]**, Wencheng Zhang**[1,*]**, Shi Yuan**[1,*]**, Hongle Guo**[2]**, Wanghu Chen**[1,†]

[1]College of Computer Science & Engineering, Northwest Normal University
[2]School of Management, Northwest Normal University

`2023222197@nwnu.edu.cn, 202421162217@nwnu.edu.cn, 2023222147@nwnu.edu.cn,`
`guohongleself@nwnu.edu.cn, chenwh@nwnu.edu.cn`

## Abstract

Unsupervised anomaly detection of multivariate time series remains challenging in complex non-stationary dynamics, due to the high false-positive rates and limited interpretability. We propose PhysDiff, combining physics-guided decomposition with diffusion-based reconstruction, to address these issues. The physics-guided signal decomposition is introduced to disentangle overlapping dynamics by isolating high frequency oscillations and low frequency trends, which can reduce interference and provide meaningful physical priors. The reconstruction through conditional diffusion modeling captures deviations from learned normal behavior, making anomalies more distinguishable. Notably, PhysDiff introduces an amplitude-sensitive permutation entropy criterion to adaptively determine the optimal decomposition depth, and automatically extract adaptive frequency components used as explicit physics-based constraints for the diffusion process. Furthermore, the proposed conditional diffusion network employs a dual-path conditioning mechanism that integrates high-frequency and low-frequency physical priors, dynamically regulating the denoising process via a novel time frequency energy routing mechanism. By weighting reconstruction errors across frequency bands, our method improves anomaly localization and enhances interpretability. Extensive experiments on five benchmark datasets and two NeurIPS-TS scenarios demonstrate that PhysDiff outperforms 18 state-of-the-art baselines, with average F1 score improvements on both standard and challenging datasets. Experimental results validate the advantages of combining principled signal decomposition with diffusion-based reconstruction for robust, interpretable anomaly detection in complex dynamic systems.

## 1 Introduction

Multivariate time series anomaly detection is a critical task across various industries, including industrial manufacturing, financial risk management, and healthcare. Anomalies in time series data are typically categorized into point anomalies and pattern anomalies [1]. Point anomalies can be further divided into contextual and global anomalies, while pattern anomalies are classified into seasonal, shapelet, and trend anomalies based on behavior-driven taxonomies [1]. Reconstruction-based methods have demonstrated promising performance in detecting point anomalies, as these anomalies manifest as individual data points exhibiting significant deviations from the expected probability distribution [2]. However, pattern anomalies, which often consist of subtle structural changes over longer periods and remain within the range of normal values, pose a greater detection

---

[*]These authors contributed equally to this work.
[†]Corresponding author.

39th Conference on Neural Information Processing Systems (NeurIPS 2025).

challenge. In real-world scenarios, time series data is predominantly non-stationary, and genuine anomalies frequently manifest as pattern anomalies. Although considerable progress has been made, most existing research focuses primarily on stationary datasets. As a result, when point anomalies are obscured, the effectiveness of conventional methods declines significantly.

Diffusion models have emerged as a powerful approach for time series anomaly detection due to their unique ability to model complex data distributions through a progressive denoising process. Unlike traditional reconstruction methods based on simple encoder-decoder architectures, diffusion models demonstrate a stronger ability to model complex patterns and temporal dependencies in non-stationary time series data. This advantage stems from their iterative noise-adding and denoising procedures [3]. This progressive approach makes diffusion models particularly effective at reconstructing normal patterns while amplifying the reconstruction error for anomalous segments. Furthermore, the probabilistic nature of diffusion models enables them to quantify uncertainty in predictions, providing valuable information for distinguishing between normal variations and genuine anomalies. By learning the distribution of normal time series behaviors, diffusion models can generate counterfactual normal versions of input sequences, making them ideal for detecting both point anomalies and subtle pattern anomalies that conventional methods often miss in non-stationary environments. To address non-stationarity problem, conditional diffusion models enhance generation capabilities through contextual information integration. These models rely on conditional decomposition to identify dynamic changes between signals and use constrained reconstruction to regulate the denoising process accordingly.

Nevertheless, two challenges remain in applying diffusion models to time series anomaly detection. **Challenge 1:** When extracting features through conditional decomposition, transforming time series to the frequency domain makes pattern anomalies easier to detect [2], but existing frequency domain decomposition methods rely on manually preset decomposition levels or energy thresholds to determine termination conditions [4]. Such static criteria are ill-suited for complex, real-world non-stationary data, where adaptive decomposition is necessary. **Challenge 2:** In the process of constrained sequence reconstruction, the reasonable use of input conditions for denoising and enhancing model interpretability is rarely mentioned. The original sequence, when decomposed, produces multiple detail coefficient sequences [5], representing signal information at various frequency levels. High frequency components typically capture transient, non-stationary changes with high uncertainty and entropy, while low frequency components reflect more stable trends with lower entropy. Overemphasis on low frequency information can obscure anomalies, whereas excessive incorporation of high frequency details risks model overfitting. These challenges highlight the need for a more dynamic and physically-informed approach to decomposition and reconstruction.

Inspired by the concept of entropy in physics, we propose PhysDiff, a physically-guided diffusion model for multivariate time series anomaly detection that combines permutation entropy with an amplitude-sensitive weighting scheme based on actual magnitude variations [6]. For Challenge 1, we dynamically adjust the decomposition depth during frequency-domain analysis by leveraging entropy values that reflect the intrinsic physical characteristics of the data. Specifically, signal complexity is assessed using an amplitude-weighted entropy measure, which effectively captures both the average magnitude and the variability of the sequence. By monitoring the entropy of residual components after each decomposition layer, the stopping condition becomes data-driven rather than heuristic. For Challenge 2, we propose an innovative reconstruction strategy based on a conditional diffusion model, in which amplitude-weighted entropy acts as a dynamic constraint during the generative reconstruction process. This approach leverages the controllability of conditional diffusion models to dynamically regulate the reconstruction of each frequency component.

The main contributions of this article are as follows:

- We propose a physically-guided diffusion model that effectively addresses non-stationarity challenges.

- We introduce an amplitude-sensitive permutation entropy guided decomposition mechanism that dynamically determines optimal decomposition depth.

- We develop a dual-path conditional diffusion framework with a novel frequency-based routing attention mechanism.

- Extensive experimental validation demonstrates that the method surpasses 18 state-of-the-art baselines, with improved average F1 score on both standard benchmarks and challenging NeurIPS-TS datasets.

## 2 Related Work

Anomaly detection of multivariate time series, particularly on large-scale non-stationary datasets, remains a significant and widely recognized challenge. According to detection strategies, existing approaches can be categorized into forecasting-based, reconstruction-based, distance-based, encoding-based, distribution-based, and tree-based ones. They offer distinct advantages but also face various challenges when addressing the complexities of real-world time series data, especially under non-stationary conditions where both the statistical properties and underlying patterns may evolve over time.

Forecasting-based methods [7] construct predictive models from historical data but struggle to distinguish anomalies from normal fluctuations due to inherent data instability. Distance-based methods [8, 9] compute similarities between subsequences yet typically fail to capture joint multivariate anomalies when they treat variables independently or use naive concatenation. Encoding-based approaches [10, 11] transform subsequences into discrete symbols or probabilistic representations, but discretization can obscure fine-grained variations critical for detecting non-stationary fluctuations. Distribution-based methods [12, 13] assume specific statistical distributions, rendering them sensitive to extreme values while neglecting temporal dependencies that are essential for modeling complex multivariate structures. Tree-based methods [14] segment data using isolation trees but overlook temporal continuity and dynamic dependencies by treating time series as independent points, significantly hampering contextual anomaly detection.

Reconstruction-based approaches [15] have emerged as particularly promising, employing encoder-decoder architectures to extract essential data representations with robust performance across various domains. A significant advancement in this category came with the introduction of Denoising Diffusion Probabilistic Models [3], which reformulate the conventional encoding-decoding paradigm into progressive diffusion and reconstruction processes. These models operate by gradually adding noise to data and learning to reverse this process, providing a powerful framework for capturing complex distributions and temporal dependencies in non-stationary time series [16]. However, current diffusion-based methods still face two critical limitations: rigid decomposition processes with static parameters that cannot adapt to evolving time series characteristics, and reconstruction strategies lacking fine-grained control over the integration of high and low frequency information. These shortcomings often result in either underfitting (missing subtle anomalies) or overfitting (generating false positives), motivating our development of PhysDiff as a physically-informed approach that dynamically adapts to complex non-stationary behaviors in multivariate time series.

## 3 Methodology

### 3.1 Framework Overview

Given a multivariate time series:

$$X = \{x_1, x_2, \ldots, x_T\} \in \mathbb{R}^{T \times d}$$

where $T$ denotes the sequence length and $d$ is the feature dimensionality, our goal is to identify anomalous patterns that deviate from typical behaviors. To facilitate local context modeling, we employ a **sliding-window paradigm** that partitions the original series into overlapping windows:

$$W = \{W_1, W_2, \ldots, W_{T-w+1}\}$$

where, each window $W_i \in \mathbb{R}^{w \times d}$ contains $w$ consecutive observations. The proposed **PhysDiff** framework leverages diffusion models as a foundation for anomaly detection due to their exceptional ability to learn complex data distributions and reconstruct normal patterns. As depicted in Figure 1, our framework consists of three key modules: **(1) Physics-Guided Feature Extraction**, where we extract interpretable features via adaptive multi-scale signal decomposition, capturing both transient dynamics and long-term trends that serve as physical priors for conditioning the subsequent diffusion process; **(2) Physically-Informed Diffusion Model**, which employs a conditional generative diffusion

process incorporating these physical priors to robustly learn the distribution of normal patterns; and **(3) Anomaly Detection Scoring Module**, where anomalies are detected by contrasting reconstructed signals against observed data, with mechanisms sensitive to both point anomalies and sequence-level pattern deviations.

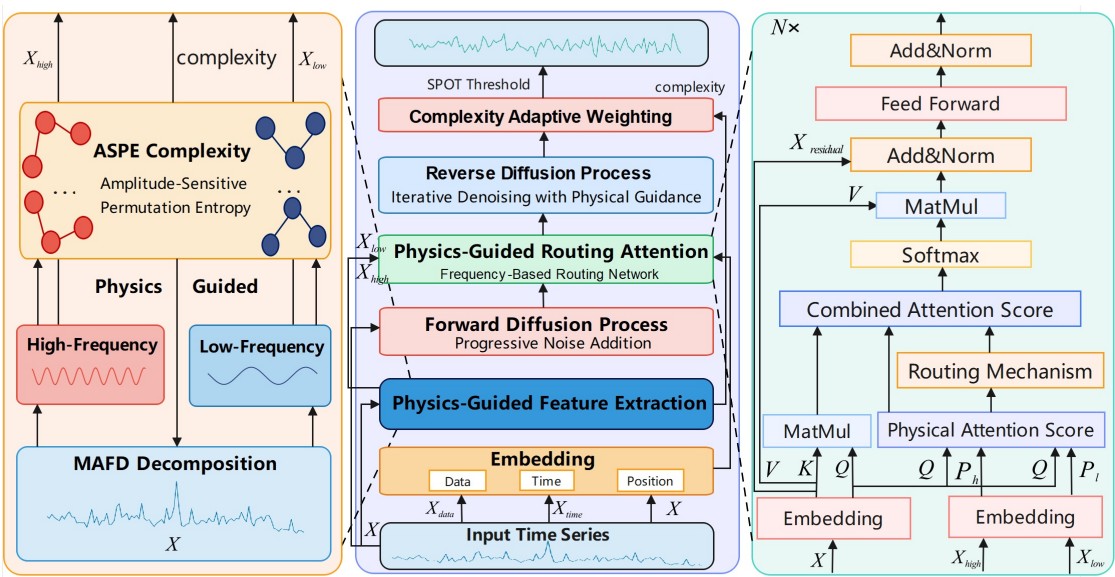

Figure 1: A physically-guided decomposition and diffusion reconstruction framework (PhysDiff). The framework contains three main modules: (1) Physics-Guided Feature Extraction; (2) Physically-Informed Diffusion Model; (3) Physics-Driven Anomaly Detection.

## 3.2 Diffusion-Based Anomaly Detection Foundations

The forward process incrementally adds noise to the original data according to a predefined schedule, transforming a clean signal into pure noise:

$$q(\mathbf{x}_{1:T}|\mathbf{x}_0) = \prod_{t=1}^{T} q(\mathbf{x}_t|\mathbf{x}_{t-1}), \quad q(\mathbf{x}_t|\mathbf{x}_{t-1}) = \mathcal{N}(\sqrt{1-\beta_t}\mathbf{x}_{t-1}, \beta_t\mathbf{I}) \tag{1}$$

where $\beta_t$ controls the noise schedule at each diffusion step. The reverse process then iteratively denoises the signal by learning a sequence of transformations:

$$p_\theta(\mathbf{x}_{0:T}) = p(\mathbf{x}_T)\prod_{t=1}^{T} p_\theta(\mathbf{x}_{t-1}|\mathbf{x}_t) \tag{2}$$

For anomaly detection, we leverage this framework by training the model exclusively on normal patterns. During inference, anomalous sequences will result in higher reconstruction errors as the model attempts to normalize them according to learned patterns. This approach is particularly effective for detecting subtle pattern anomalies in non-stationary environments.

## 3.3 Physics-Guided Feature Extraction

To enhance the diffusion model's ability to capture complex time series dynamics, a physics-guided feature extraction method is proposed. This method adaptively decomposes multivariate signals into interpretable components and quantitatively assesses signal complexity. These extracted physical features serve as conditioning information to guide the subsequent diffusion-based reconstruction, enabling the model to distinguish normal variations from genuine anomalies.

Our approach generalizes adaptive Fourier decomposition principles [17] to the multivariate scenario via a novel Multi-channel Adaptive Fourier Decomposition (MAFD). Specifically, for a multivariate

time series $\mathbf{X}(t) = [x_1(t), x_2(t), \ldots, x_C(t)]$, MAFD recursively decomposes the signal using cross-channel shared Blaschke basis functions:

$$\mathbf{X}(t) = \sum_{n=1}^{N} \mathbf{A}_n \cdot B_n(e^{jt}) + \mathbf{R}_N(t) \tag{3}$$

where the projection coefficients are defined as $\mathbf{A}_n = [A_{1,n}, A_{2,n}, \ldots, A_{C,n}]$, and $\mathbf{R}_N(t)$ denotes the residual component. The basis function selection maximizes cross-channel energy convergence: $\max_{a_n} \sum_{c=1}^{C} |\langle G_c, B_n \rangle|^2$.

A critical challenge in adaptive decomposition is determining the optimal stopping point that balances signal structure preservation with anomaly distinguishability. To address this, we introduce an amplitude-sensitive permutation entropy (ASPE) measure [6] that dynamically determines decomposition depth:

$$H_{\text{ASPE}} = -\frac{1}{\ln(d!)} \sum_{\pi} \omega_\pi P(\pi) \ln P(\pi) \tag{4}$$

where $P(\pi)$ is the probability of ordering pattern $\pi$, and the weight $\omega_\pi = \frac{\sigma(y_\pi^{d,\tau})}{\mu(y_\pi^{d,\tau})}$ emphasizes amplitude variations. ASPE effectively discriminates between meaningful signal structure and noise by monitoring the complexity of residual components at each decomposition layer. When ASPE stabilizes and stops decreasing significantly, this indicates that the residual is dominated by unstructured variability rather than meaningful patterns. This data-driven stopping criterion ensures that decomposition proceeds just enough to isolate physically interpretable frequency components without over-segmenting the signal or burying anomaly-relevant features in irrelevant high-frequency bands.

### 3.4 Physically-Informed Diffusion Model

#### 3.4.1 Conditional Diffusion with Physical Priors

In PhysDiff, the basic diffusion framework is extended by conditioning the reverse process on physical priors extracted from the data as:

$$p_\theta(\mathbf{x}_{0:T}|z_t) = p(\mathbf{x}_T) \prod_{t=1}^{T} p_\theta(\mathbf{x}_{t-1}|\mathbf{x}_t, z_t) \tag{5}$$

where the physical context information $z_t$ is defined as a composite feature vector:

$$z_t = \{P_h(t), P_l(t), H_{\text{ASPE}}(t), E(\mathbf{x}_t)\} \tag{6}$$

Here, $P_h(t)$ and $P_l(t)$ represent high-frequency and low-frequency components obtained from MAFD decomposition, capturing rapid transient changes and long-term trends respectively; $H_{\text{ASPE}}(t)$ is the amplitude-sensitive permutation entropy value quantifying signal complexity; and $E(\mathbf{x}_t)$ is the physics-based energy function measuring physical plausibility of the current state. Notably, these components serve as conditioning features that guide the denoising process, rather than being the targets of diffusion themselves. This conditioning mechanism allows the model to incorporate domain knowledge about time series dynamics, significantly improving its ability to distinguish between normal variations and anomalous patterns.

Furthermore, to integrate MAFD and ASPE into the conditioning mechanism, the amplitude-sensitive weights are directly embedded into the basis selection process as follows:

$$\mathcal{L}_{\text{MAFD-ASPE}} = \sum_{c=1}^{C} \left( \|G_c - \sum_{n=1}^{N} A_{c,n} B_n\|^2 + \lambda H_{\text{ASPE}} \right) \tag{7}$$

This composite loss not only refines the decomposition strategy but also increases sensitivity to abnormal patterns while preserving physical interpretability.

#### 3.4.2 Frequency-Based Routing Attention

Notably, PhysDiff introduces a cross-modal routing mechanism that enables dynamic interaction between high frequency and low frequency components during the diffusion process. Specifically, the

input signal $\mathbf{X}(t)$ is first decomposed via MAFD into high-frequency and low-frequency components: $\mathbf{X}(t) = \mathbf{X}_{\text{HF}}(t) + \mathbf{X}_{\text{LF}}(t)$, where $\mathbf{X}_{\text{HF}}$ captures rapid transients and $\mathbf{X}_{\text{LF}}$ represents stable trends. These physical components then serve as conditioning information for the routing network:

$$z_t = \mathcal{R}(t, \mathbf{X}_{\text{HF}}(t), \mathbf{X}_{\text{LF}}(t)) \tag{8}$$

It is important to note that the diffusion process operates on the original time series $\mathbf{X}(t)$, while $\mathbf{X}_{\text{HF}}(t)$ and $\mathbf{X}_{\text{LF}}(t)$ guide the reconstruction through the routing attention mechanism.

In detail, the physically guided attention is computed by:

$$\text{Attention}(Q, K, V, \mathbf{P}_h, \mathbf{P}_l) = \text{softmax}\left(\frac{QK^T + g_h \cdot Q\mathbf{P}_h^T + g_l \cdot Q\mathbf{P}_l^T}{\sqrt{d_k}}\right) V \tag{9}$$

with adaptive gating coefficients $g_h = \sigma(Q\mathbf{P}_h^T)$ and $g_l = \sigma(Q\mathbf{P}_l^T)$. This mechanism allows the model to dynamically emphasize either rapid transient changes or stable trend information according to the contextual requirements at each diffusion step.

### 3.4.3 Physical Consistency through Energy Guidance

To ensure that generated samples adhere to underlying physical principles, we introduce an energy function derived from instantaneous frequency analyses:

$$\mathcal{E}(\mathbf{x}_t) = \|\nabla_{\mathbf{x}}\Phi(\mathbf{x}_t)\|^2 \tag{10}$$

where $\Phi(\cdot)$ represents the physical potential field learned from normal patterns. This energy term regularizes the denoising process by penalizing physically implausible states.

Moreover, physical consistency is enhanced by integrating Langevin dynamics into the sampling procedure as follows:

$$\mathbf{x}_{t-1} \leftarrow \mathbf{x}_t - \lambda\nabla_{\mathbf{x}_t}\mathcal{E}(\mathbf{x}_t) + \sqrt{2\lambda}\mathbf{n} \tag{11}$$

where $\lambda$ represents the step size and $\mathbf{n}$ denotes Gaussian noise. Additionally, a frequency-adaptive dynamic trend influence factor $\gamma(t, \omega) = \sigma(10(1 - \sqrt{\bar{\alpha}_t}) \cdot A_\omega)$ modulates the reliance on physical components throughout the denoising trajectory, where $A_\omega = \exp(-|\omega|^2/\sigma^2)$ is the frequency response function. This design allows different frequency bands to receive adaptive physical guidance strength at different diffusion stages, ensuring stronger physical guidance when needed most while allowing the model to capture fine-grained details that may not be explicitly encoded in the physical priors.

## 3.5 Training Strategy and Optimization

The entire framework is trained by minimizing a composite loss function:

$$\mathcal{L}_{\text{total}} = \mathcal{L}_{\text{diff}} + \gamma \cdot \mathcal{L}_{\text{MAFD-ASPE}} + \eta \cdot \mathcal{E}(\mathbf{x}_t) \tag{12}$$

where the diffusion loss $\mathcal{L}_{\text{diff}} = \mathbb{E}_{t,\mathbf{x}_0,\epsilon}\left[\|\epsilon - \epsilon_\theta(\mathbf{x}_t, t, z_t)\|^2\right]$ ensures accurate modeling of normal patterns, $\mathcal{L}_{\text{MAFD-ASPE}}$ incorporates physical insights, and $\mathcal{E}(\mathbf{x}_t)$ enforces physical plausibility.

To improve robustness, we incorporate a controlled disturbance mechanism during training where $\mathbf{W}_i^{\text{disturbed}} = \mathbf{W}_i + \delta \cdot p$, with $\delta \sim \mathcal{U}(0, 1)$ as a random perturbation. Optimization uses the AdamW optimizer with early stopping to mitigate overfitting.

## 3.6 Physics-Driven Anomaly Detection

For each sliding window, we define an anomaly score that fuses reconstruction fidelity with physical plausibility:

$$\text{Score}(t) = \alpha \cdot \underbrace{\|\mathbf{X}(t) - \hat{\mathbf{X}}(t)\|_2}_{\text{Reconstruction Error}} + (1 - \alpha) \cdot \underbrace{D_{\text{KL}}(P_{\text{TTFD}}\|P_{\text{prior}})}_{\text{Time-Frequency Distribution Divergence}} \tag{13}$$

where $\hat{\mathbf{X}}(t)$ is the reconstruction of the observed window, $P_{\text{TTFD}}$ represents the transient time-frequency distribution, and $P_{\text{prior}}$ denotes the prior distribution of normal patterns.

We employ extreme value theory through SPOT [18] to dynamically calibrate the detection threshold by fitting a generalized Pareto distribution to the tail of the anomaly score distribution, deriving the threshold as $\tau_t = \mu_{\text{score}} + k \cdot \sigma_{\text{score}}$. This adaptive thresholding strategy accommodates data-specific characteristics, enabling robust anomaly detection across diverse time series domains.

# 4 Experiments

## 4.1 Experimental Details

**Datasets.** We evaluated our approach using five widely recognized benchmark datasets: SMD [19], MSL [7], SMAP [7], SWaT [20], and PSM [21], plus the NeurIPS-TS dataset—comprising Creditcard and GECCO subsets as detailed by Lai et al. (2021) [1]. Data labeled as normal were partitioned with 80% allocated for training and 20% for validation, ensuring the model is properly optimized on typical behavior. These datasets represent diverse domains including spacecraft telemetry, water treatment systems, and financial transactions, providing a comprehensive evaluation landscape for anomaly detection methods.

**Metrics.** The affiliation-based F1 score is adopted as our primary performance metric, which comprised precision and recall indicators extended from affiliation metrics [22]. Traditional point adjustment (PA) methods, where detecting a single point within an anomalous segment counts as successful detection of the entire segment, can lead to overly optimistic evaluations by inflating true positives and suppressing false negatives [11, 16]. The affiliation-based F1 score offers a more rigorous evaluation by computing precision and recall based on the average directed distance between predicted anomaly events and ground truth, accounting for both spatial and temporal adjacency. All results are reported as percentages, with best performances in bold and second-best underlined.

**Baselines.** Our method is compared against 18 state-of-the-art anomaly detection approaches across six categories: (1) Forecasting methods (LSTM [7]); (2) Reconstruction methods (PCA [23], AE [24], DAGMM [25], BeatGAN [15], OmniAnomaly [19], D3R [16]); (3) Distance methods (OCSVM [26], HBOS [13], LOF [9], DeepSVDD [12]); (4) Encoding methods (Anomaly Transformer [27], DCdetector [11], SensitiveHUE [28], MTAD-GAT [29], TFAD [10]); (5) Distribution methods (LODA [30]); (6) Tree methods (iForest [14]). Additionally, we included an adversarial baseline marking anomalies at fixed intervals to provide a non-informative temporal reference. For baselines with publicly available implementations, we re-ran experiments using official code and recommended hyperparameters. For methods where identical results on the same datasets with the same evaluation protocol were reported in recent work [11, 16], we directly cite those results to ensure methodological consistency and fair comparison.

## 4.2 Comparative Study

Our comprehensive comparative analysis evaluates PhysDiff over 18 state-of-the-art anomaly detection methods across five real-world benchmark datasets, as shown in Table 1. PhysDiff consistently

Table 1: Performance comparison on five real-world anomaly detection datasets.

| Dataset | SMD | | | MSL | | | SMAP | | | SWaT | | | PSM | | |
|---|---|---|---|---|---|---|---|---|---|---|---|---|---|---|---|
| Metric | P | R | F1 | P | R | F1 | P | R | F1 | P | R | F1 | P | R | F1 |
| OCSVM | 66.98 | 62.03 | 73.75 | 50.26 | 99.86 | 66.87 | 41.05 | 69.37 | 51.58 | 56.08 | 98.72 | 72.11 | 57.51 | 58.11 | 57.81 |
| PCA | 64.92 | 40.19 | 54.34 | 52.69 | 98.33 | 68.61 | 50.62 | 98.48 | 66.87 | 62.32 | 82.96 | 71.18 | 77.44 | 37.71 | 53.53 |
| HBOS | 56.28 | 63.11 | 62.17 | 59.25 | 83.32 | 69.25 | 41.54 | 66.17 | 51.04 | 57.71 | 29.82 | 43.21 | 100.00 | 6.54 | 12.28 |
| LOF | 57.69 | 99.10 | 72.92 | 49.89 | 72.18 | 59.00 | 47.92 | 82.86 | 60.72 | 53.20 | 96.73 | 68.65 | 53.90 | 99.91 | 70.02 |
| IForset | 100.00 | 9.37 | 17.13 | 53.87 | 94.58 | 68.65 | 41.12 | 68.91 | 51.51 | 53.03 | 62.80 | 62.03 | 100.00 | 3.35 | 6.48 |
| LODA | 59.02 | 66.18 | 62.40 | 57.79 | 95.65 | _72.05_ | 51.51 | 100.00 | 68.00 | 56.30 | 70.14 | 62.54 | 62.22 | 40.17 | 56.05 |
| AE | 69.22 | 98.48 | _81.30_ | 55.75 | 96.66 | 70.72 | 39.42 | 70.31 | 50.52 | 54.92 | 98.20 | 70.45 | 60.67 | 98.24 | _75.01_ |
| DAGMM | 63.57 | 70.83 | 67.00 | 54.07 | 92.11 | 68.14 | 50.75 | 96.38 | 66.49 | 59.42 | 92.36 | _72.32_ | 68.22 | 70.50 | 69.34 |
| LSTM | 60.12 | 84.77 | 70.35 | 58.82 | 14.68 | 23.49 | 55.25 | 27.70 | 36.90 | 49.99 | 82.11 | 62.15 | 57.06 | 95.92 | 71.55 |
| BeatGAN | 74.11 | 81.64 | 77.69 | 55.74 | 98.94 | 71.30 | 54.04 | 98.30 | 69.74 | 61.89 | 83.46 | 71.08 | 58.81 | 99.08 | 73.81 |
| Omni | 79.09 | 75.77 | 77.40 | 51.23 | 99.40 | 67.61 | 52.74 | 98.37 | 68.70 | 62.76 | 82.82 | 71.41 | 69.20 | 80.79 | 74.55 |
| A.T. | 100.00 | 3.19 | 6.19 | 51.04 | 95.36 | 66.49 | 56.91 | 96.69 | _71.65_ | 53.63 | 59.94 | 57.59 | 52.01 | 82.18 | 64.55 |
| DCdetector | 50.93 | 95.57 | 66.45 | 55.94 | 95.53 | 70.56 | 53.12 | 98.37 | 68.99 | 53.25 | 98.12 | 69.03 | 54.72 | 86.36 | 66.99 |
| SensitiveHUE | 60.34 | 90.13 | 72.29 | 55.92 | 98.95 | 71.46 | 53.63 | 98.37 | 69.42 | 58.91 | 91.71 | 71.74 | 56.15 | 98.75 | 71.59 |
| DeepSVDD | 64.98 | 64.77 | 64.88 | 10.53 | 100.00 | 19.06 | 29.73 | 7.09 | 11.45 | 59.11 | 93.53 | 72.44 | 74.05 | 50.64 | 60.15 |
| MTAD-GAT | 85.90 | 67.69 | 75.71 | 54.96 | 94.93 | 69.81 | 39.05 | 93.99 | 55.08 | 65.90 | 77.51 | 71.23 | 79.90 | 60.14 | 68.63 |
| TFAD | 56.32 | 97.83 | 71.49 | 54.96 | 94.93 | 69.81 | 39.05 | 93.99 | 55.08 | 81.96 | 69.53 | 60.38 | 79.14 | 71.63 | **75.20** |
| D3R | 64.87 | 97.93 | 78.04 | 56.45 | 95.55 | 71.81 | 51.08 | 94.46 | 66.30 | 64.25 | 77.50 | 70.25 | 53.17 | 100.00 | 69.43 |
| **PhysDiff** | 71.03 | 96.00 | **81.65** | 62.75 | 92.66 | **74.83** | 64.36 | 86.81 | **73.91** | 60.00 | 92.04 | **72.64** | 66.09 | 84.47 | 74.16 |

outperforms all baseline models, most notably achieving an F1 score of 74.16% on PSM, outperforming AE by over 7 percentage points, and 81.65% on SMD, surpassing the previous best AE performance of 81.30%. On datasets characterized by complex temporal dynamics, the improvements are particularly significant. PhysDiff achieves 73.91% F1 score on SMAP, outperforming Anomaly Transformer by more than 2 percentage points, and 74.83% on MSL, exceeding LODA by 2.78

percentage points. These gains can be attributed to our model's ability to effectively decompose time series into physically meaningful components using the proposed Multi-channel Adaptive Fourier Decomposition.

To validate PhysDiff's performance in more challenging scenarios, we conducted additional experiments on the NeurIPS-TS datasets featuring more complex anomaly patterns and higher-dimensional feature spaces. As shown in Table 2, PhysDiff achieves F1 scores of 69.44% and 57.92% on the Creditcard and GECCO datasets respectively, substantially outperforming previous best approaches. On Creditcard, PhysDiff surpasses D3R by 6.82 percentage points, while on GECCO it exceeds Anomaly Transformer by 1.96 percentage points. These significant improvements demonstrate our approach's superior ability to detect anomalies in complex real-world scenarios.

The consistent performance gains across all seven datasets highlight several key advantages of our physically-guided diffusion approach. The adaptive decomposition mechanism effectively captures both high frequency transient patterns and low frequency trend components, enabling more accurate anomaly detection across different anomaly types. The incorporation of amplitude-sensitive permutation entropy provides a physically meaningful measure of signal complexity that helps distinguish between normal variations and

Table 2: Performance comparison on the NeurIPS-TS datasets.

| Dataset | Creditcard | | | GECCO | | |
|---|---|---|---|---|---|---|
| Metric | P | R | F1 | P | R | F1 |
| iForst | 1.52 | 88.01 | 2.99 | 2.59 | 97.40 | 5.04 |
| AE | 14.76 | 31.71 | 20.14 | 76.56 | 20.14 | 31.89 |
| A.T. | 55.86 | 57.55 | 56.69 | 54.77 | 57.21 | _55.96_ |
| DCdetector | 49.86 | 67.67 | 57.41 | 38.30 | 59.70 | 46.60 |
| D3R | 56.90 | 69.61 | _62.62_ | 59.00 | 36.61 | 45.18 |
| **PhysDiff** | 64.10 | 75.88 | **69.44** | 60.89 | 55.23 | **57.92** |

actual anomalies. Our frequency-adaptive routing mechanism dynamically calibrates the contributions of different frequency components during the diffusion process, leading to more accurate reconstruction of normal patterns. Quantitative analysis confirms PhysDiff's superior effectiveness on non-stationary datasets compared to stationary ones. Datasets exhibiting strong non-stationarity characteristics achieve larger performance improvements, with detailed statistical validation provided in Appendix E.3. This validates our hypothesis that adaptive decomposition mechanisms are particularly beneficial for handling complex temporal patterns in non-stationary environments. These results collectively establish PhysDiff as a new state-of-the-art in multivariate time series anomaly detection, offering both superior performance and enhanced interpretability through its physics-guided approach.

## 4.3 Ablation Study

To evaluate each component's contribution to our PhysDiff framework, we conducted comprehensive ablation studies across five datasets in Table 3. The physical guidance mechanism proved most critical, with its removal causing the largest performance drop of 5.95% average F1 score, confirming our hypothesis that incorporating physical constraints substantially improves anomaly detection accuracy. Similarly, disabling routing attention and high frequency components led to significant performance degradation of 10.20% and 6.63% respectively, with particularly severe impacts on specific datasets: routing attention removal caused 25.13% decrease on PSM while high frequency component removal resulted in 11.14% decrease on SMAP. Our information-theoretic components also proved essential, as removing permutation entropy caused a 5.32% average decrease, while replacing amplitude-sensitive permutation entropy with standard entropy resulted in a 4.38% reduction.

Table 3: Results of ablation studies. F1 scores are reported, with higher values meaning better performance. The best scores are highlighted in bold.

| dataset | PhysDiff | w/o PE | w/o ASPE | w/o RA | w/o HF | w/o LF | w/o PG | w/o coms(8) | w/o coms(16) |
|---|---|---|---|---|---|---|---|---|---|
| SMD | **81.65** | 70.61 | 70.51 | 72.37 | 70.91 | 70.02 | 70.84 | 69.86 | 69.49 |
| MSL | **74.83** | 71.59 | 71.06 | 67.95 | 69.92 | 71.46 | 70.54 | 71.59 | 71.12 |
| SMAP | **73.91** | 70.76 | 70.51 | 66.40 | 62.77 | 67.27 | 65.03 | 65.70 | 68.78 |
| SWaT | **72.64** | 68.76 | 69.84 | 70.46 | 69.81 | 69.84 | 70.41 | 69.84 | 69.83 |
| PSM | **74.16** | 68.86 | 73.38 | 49.03 | 70.65 | 68.86 | 70.62 | 73.47 | 68.86 |
| Average | **75.44** | 70.12 | 71.06 | 65.24 | 68.81 | 69.49 | 69.49 | 70.09 | 69.62 |

Regarding decomposition granularity, neither 8 nor 16 components achieved optimal performance, showing decreases of 5.35% and 5.82% respectively, demonstrating the importance of appropriate decomposition depth selection. Low frequency component removal caused a 5.95% decrease, with varying impacts across datasets. These findings collectively confirm that PhysDiff's superior performance stems from the synergistic integration of physical insights, adaptive frequency decomposition, and information-theoretic measures. Each component makes substantial contributions to the framework's effectiveness across diverse real-world scenarios, validating our architectural design choices and providing empirical evidence for the importance of incorporating domain knowledge into anomaly detection systems.

## 4.4 Case Study: Fraud Detection in Financial Transactions

The effectiveness of PhysDiff in real-world scenarios is demonstrated through its application to financial transaction fraud detection. Our analysis of the feature importance rankings in Figure 2 reveals that Feature 1 with importance score 0.0998, Feature 15 with 0.0677, and Feature 22 with 0.0650 contribute most significantly to fraud identification, collectively accounting for over 23% of the model's discriminative power. As shown in Figure 3, Feature 1 exhibits a particularly distinctive pattern where fraudulent transactions display a wider, left-skewed distribution compared to the sharp, concentrated peaks of normal transactions. This pattern provides strong discriminative signals for the detection model.

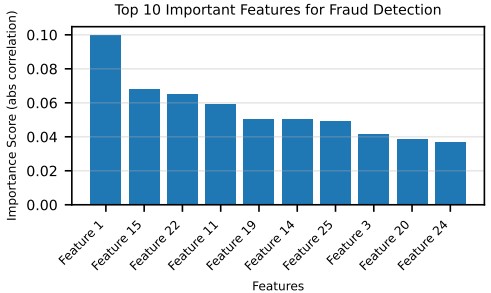
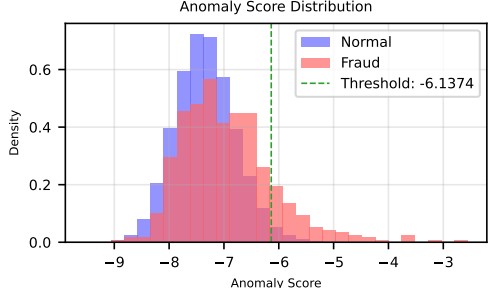

Figure 2: Top 10 features for fraud detection, with Feature 1 (0.0998) contributing most significantly.

Figure 3: Anomaly score distribution showing separation between normal transactions (below -6.1374) and fraudulent ones.

The anomaly detection timeline in Figure 4 illustrates PhysDiff's performance across 1,000 sequential transactions, where the model successfully identifies two true fraud cases while generating several false alarms at high anomaly score peaks. The anomaly score distribution shows clear separation between normal and fraudulent transactions, with most normal transactions generating scores below the determined threshold of -6.1374, while fraudulent transactions frequently produce higher scores. This separation enables the model to achieve 64.10% precision and 75.88% recall, resulting in a 69.44% F1 score that significantly outperforms traditional fraud detection approaches.

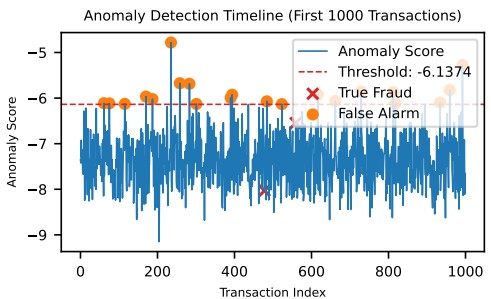
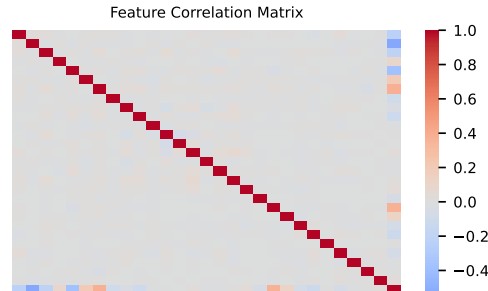

Figure 4: Detection timeline across 1,000 transactions identifying true fraud cases (red x) and false alarms (orange).

Figure 5: Feature correlation matrix revealing inter-feature relationships that help identify suspicious activities.

The feature correlation matrix in Figure 5 and distribution analyses reveal that PhysDiff effectively captures complex inter-feature relationships that would be missed by univariate detection methods. Features with lower individual importance rankings show distinct separation patterns in their distributions that complement the primary features. This multi-scale decomposition enables PhysDiff to distinguish between natural transaction variability and truly suspicious patterns, making it particularly effective at identifying complex fraud schemes that evolve gradually across multiple transactions. This represents a significant advantage over conventional threshold-based detection systems.

## 5  Conclusion

We proposed **PhysDiff**, a physically-guided decomposition and diffusion framework for anomaly detection in non-stationary multivariate time series. By combining amplitude-sensitive permutation entropy for optimized decomposition with dual-path conditional diffusion, our approach effectively models both abrupt failures and gradual degradations while maintaining interpretability. Experiments show that PhysDiff achieves improved average F1 score on both standard and NeurIPS-TS datasets, outperforming state-of-the-art methods. Its effectiveness across domains from industrial systems to financial fraud detection validates the integration of physical priors into deep generative models. Future work includes improving computational efficiency, incorporating privacy mechanisms, and extending the framework to handle multimodal data with varying sampling rates and missing values.

## Acknowledgements

This work was supported by the National Natural Science Foundation of China (Grant No. 62462059).

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

# Appendix / supplemental material

## A  Notations

In this paper, we employ various mathematical notations to represent the components of our proposed PhysDiff framework. To facilitate the reader's understanding of the mathematical formulations presented throughout the paper, we provide a comprehensive list of the key notations in Table 4. These notations cover the multivariate time series representation, decomposition components, diffusion model parameters, and anomaly detection metrics. Understanding these notations is essential for following the theoretical development and algorithmic implementations of our physically-guided approach to time series anomaly detection. The reader is encouraged to refer to this table when interpreting the equations and algorithms presented in subsequent sections.

Table 4: Mathematical Notations Used in PhysDiff

| Symbol | Description |
| --- | --- |
| $X = \{x_1, x_2, \ldots, x_T\} \in \mathbb{R}^{T \times d}$ | Multivariate time series data with $T$ sequence length and $d$ dimensions |
| $W = \{W_1, W_2, \ldots, W_{T-w+1}\}$ | Sequence of overlapping windows with $W_i \in \mathbb{R}^{w \times d}$ |
| $X(t) = [x_1(t), x_2(t), \ldots, x_C(t)]$ | Multivariate time series with $C$ channels |
| $B_n(e^{jt})$ | Blaschke basis functions in MAFD decomposition |
| $A_n = [A_{1,n}, A_{2,n}, \ldots, A_{C,n}]$ | Projection coefficients in MAFD |
| $R_N(t)$ | Residual component after decomposition |
| $H_{\text{ASPE}}$ | Amplitude-sensitive permutation entropy |
| $P(\pi)$ | Probability of permutation pattern $\pi$ |
| $\omega_\pi = \frac{\sigma(y_\pi^{d,\tau})}{\mu(y_\pi^{d,\tau})}$ | Weight for amplitude variations |
| $X_{\text{HF}}(t), X_{\text{LF}}(t)$ | High-frequency and low-frequency components |
| $q(x_{1:T}|x_0)$ | Forward diffusion process |
| $p_\theta(x_{0:T}|z_t)$ | Reverse diffusion process |
| $\beta_t$ | Noise schedule parameter |
| $\alpha_t = 1 - \beta_t$ | Single step noise parameter |
| $\bar{\alpha}_t = \prod_{i=1}^{t} \alpha_i$ | Cumulative noise parameter |
| $\epsilon_\theta$ | Noise prediction network |
| $E(x_t) = \|\nabla_x \Phi(x_t)\|^2$ | Physics-based energy function |
| $\text{Score}(t)$ | Anomaly detection score |
| $\gamma(t) = \sigma(10(1 - \sqrt{\bar{\alpha}_t}))$ | Dynamic trend influence factor |
| $F_{\gamma,\sigma}(y)$ | Generalized Pareto distribution for threshold calibration |
| $\tau_t = \mu_{\text{score}} + k \cdot \sigma_{\text{score}}$ | Adaptive anomaly threshold |
| $\mathcal{L}_{\text{MAFD-ASPE}}$ | Composite loss for decomposition |
| $\mathcal{L}_{\text{diff}}$ | Diffusion model loss |
| $\mathcal{L}_{\text{total}}$ | Total framework loss |

## B  MAFD and ASPE

### B.1  Multi-channel Adaptive Fourier Decomposition

The Multi-channel Adaptive Fourier Decomposition (MAFD) [17] extends traditional adaptive Fourier decomposition to multivariate scenarios. MAFD recursively decomposes multi-channel signals in a hierarchical manner, with each level of decomposition building upon the results obtained from the previous stage. The analytic signal of the $c$-th channel is expressed as:

$$G_c\left(e^{jt}\right) = \sum_{n=1}^{N} A_{c,n} \cdot B_n\left(e^{jt}\right) + R_{c,N}\left(e^{jt}\right) \tag{14}$$

where $A_{c,n}$ is the decomposition coefficient of the $c$-th channel at the $n$-th level, $B_n(e^{jt})$ denotes the shared basis function (modified Blaschke product), and $R_{c,N}(e^{jt})$ represents the remainder term after $N$-level decomposition. The decomposition process recursively determines the coefficients $A_{c,n}$ and the basis functions $B_n(e^{jt})$ at each level.

For a multivariate time series $X(t) = [x_1(t), x_2(t), \ldots, x_C(t)]$, each channel $x_c(t)$ performs the Hilbert transform to obtain the analytic signal:

$$G_c(e^{jt}) = x_c(t) + j\mathcal{H}\{x_c(t)\} \tag{15}$$

where $\mathcal{H}\{\cdot\}$ denotes the Hilbert transform. After removing the mean value of $G_c(e^{jt})$, the initial residual is defined as:

$$R_{c,0}(e^{jt}) = G_c(e^{jt}) \tag{16}$$

For each decomposition level $n = 1, 2, \ldots, N$, the recursive procedure mainly consists of three steps:

**Search for the optimal basis function parameter.** Select the parameter $a_n \in \mathbb{D}$ by maximizing the total energy across all channels:

$$a_n = \arg\max_{a \in \mathbb{D}} \sum_{c=1}^{C} \left| \langle G_{c,n}(e^{jt}), e_{\{a\}}(e^{jt}) \rangle \right|^2 \tag{17}$$

where $e_{\{a\}}(e^{jt}) = \frac{\sqrt{1-|a|^2}}{1-\bar{a}e^{jt}}$ is the normalized evaluator function.

**Compute the decomposition coefficients.** After determining the optimal $a_n$, the decomposition coefficient $A_{c,n}$ for each channel is calculated as:

$$A_{c,n} = \langle G_{c,n}(e^{jt}), e_{\{a_n\}}(e^{jt}) \rangle \tag{18}$$

which represents the projection of the residual signal onto the basis function $e_{\{a_n\}}$.

**Update the residual.** The residual $R_{c,n}(e^{jt})$ is updated as:

$$R_{c,n}(e^{jt}) = R_{c,n-1}(e^{jt}) - A_{c,n}B_n(e^{jt}) \tag{19}$$

where the basis function $B_n(e^{jt})$, constructed as a modified Blaschke product from the parameter sequence $\{a_1, a_2, \ldots, a_n\}$, is defined by:

$$B_n(e^{jt}) = \frac{\sqrt{1-|a_n|^2}}{1-\bar{a}_n e^{jt}} \prod_{d=1}^{n-1} \frac{e^{jt} - a_d}{1 - \bar{a}_d e^{jt}} \tag{20}$$

## B.2 Amplitude-Sensitive Permutation Entropy

Amplitude-Sensitive Permutation Entropy (ASPE) [6] extends traditional permutation entropy by incorporating amplitude information. The ASPE is defined as:

$$H_{ASPE} = -\frac{1}{\ln(d!)} \sum_{\pi} \omega_\pi P(\pi) \ln P(\pi) \tag{21}$$

where $P(\pi)$ is the probability of a given ordering pattern $\pi$, and the weight $\omega_\pi = \frac{\sigma(y_\pi^{d,\tau})}{\mu(y_\pi^{d,\tau})}$ emphasizes amplitude variations. This entropy measure effectively discriminates between typical variations and anomalies by integrating both ordering and amplitude information.

For a time series $\{x_t\}_{t=1}^{T}$, we construct embedding vectors $y_j^{d,\tau} = [x_j, x_{j+\tau}, \ldots, x_{j+(d-1)\tau}]$ for $j = 1, 2, \ldots, T - (d-1)\tau$, where $d$ is the embedding dimension and $\tau$ is the time delay. Each embedding vector is then assigned a permutation pattern $\pi = (r_1, r_2, \ldots, r_d)$ based on the relative ordering of its elements, where:

$$x_{j+r_1\tau} \leq x_{j+r_2\tau} \leq \ldots \leq x_{j+r_d\tau} \tag{22}$$

The probability $P(\pi)$ of each pattern is calculated as the frequency of occurrence normalized by the total number of embedding vectors.

# C  Noise diffusion proofs

## C.1  Forward Process Derivation

The forward process in our diffusion model is defined as a Markov chain that gradually adds noise to the input:

$$q(x_{1:T}|x_0) = \prod_{t=1}^{T} q(x_t|x_{t-1}), \quad q(x_t|x_{t-1}) = \mathcal{N}(\sqrt{1-\beta_t}x_{t-1}, \beta_t\mathbf{I}) \tag{23}$$

This can be rewritten in a non-recursive form to enable direct sampling from any timestep:

$$q(x_t|x_0) = \mathcal{N}(\sqrt{\bar{\alpha}_t}x_0, (1-\bar{\alpha}_t)\mathbf{I}) \tag{24}$$

where $\alpha_t = 1 - \beta_t$ and $\bar{\alpha}_t = \prod_{i=1}^{t} \alpha_i$. This formulation allows us to express $x_t$ directly in terms of $x_0$:

$$x_t = \sqrt{\bar{\alpha}_t}x_0 + \sqrt{1-\bar{\alpha}_t}\epsilon, \quad \epsilon \sim \mathcal{N}(0, \mathbf{I}) \tag{25}$$

## C.2  Reverse Process Properties

The reverse process learns to gradually denoise the signal:

$$p_\theta(x_{0:T}|z_t) = p(x_T) \prod_{t=1}^{T} p_\theta(x_{t-1}|x_t, z_t) \tag{26}$$

where $z_t$ represents the physical guidance information. The conditional density $p_\theta(x_{t-1}|x_t, z_t)$ is modeled as a Gaussian:

$$p_\theta(x_{t-1}|x_t, z_t) = \mathcal{N}(\mu_\theta(x_t, t, z_t), \Sigma_\theta(x_t, t)) \tag{27}$$

The mean $\mu_\theta$ incorporates our physics-guided information through the denoising network $\epsilon_\theta$:

$$\mu_\theta(x_t, t, z_t) = \frac{1}{\sqrt{\alpha_t}} \left( x_t - \frac{1-\alpha_t}{\sqrt{1-\bar{\alpha}_t}}\epsilon_\theta(x_t, t, z_t) \right) \tag{28}$$

The variance follows a schedule that interpolates between fixed timestep-dependent values:

$$\Sigma_\theta(x_t, t) = \frac{1-\bar{\alpha}_{t-1}}{1-\bar{\alpha}_t}(1-\alpha_t) \tag{29}$$

Our physically-guided reverse process optimizes the evidence lower bound (ELBO):

$$\mathcal{L} = \mathbb{E}_{t,x_0,\epsilon} \left[ \|\epsilon - \epsilon_\theta(x_t, t, z_t)\|^2 \right] + \lambda \cdot \mathcal{L}_{\text{MAFD-ASPE}} + \eta \cdot \mathcal{E}(x_t) \tag{30}$$

where $\mathcal{E}(x_t) = \|\nabla_x \Phi(x_t)\|^2$ represents our physics-based energy constraint.

# D  Detailed experimental settings

## D.1  Datasets

To prove the effectiveness of our method, we evaluated six real-world datasets: (1) SMD (Server Machine Dataset) collects data from 28 servers over a period of 5 weeks, with 38 metrics monitored for each machine. (2) The PSM (Pooled Server Metrics) dataset is a signal dataset from eBay's IT systems, consisting of 26 dimensions. (3) The MSL (Mars Science Laboratory rover) and SMAP (Soil Moisture Active Passive satellite) are NASA spacecraft telemetry datasets that contain multivariate data captured from multiple telemetry channels across various entities. The MSL dataset comprises 27 entities with 55 telemetry channels per entity, while the SMAP dataset contains 55 entities with 25 telemetry channels per entity. (4) The SWaT (Secure Water Treatment) was collected from 51 sensors in a critical infrastructure system under continuous operational conditions. (5) The NeurIPS-TS (NeurIPS 2021 Time Series Benchmark) introduces a collection of five time series anomaly detection scenarios, systematically classified through behavior-driven taxonomy into distinct categories: point-global, pattern-contextual, pattern-shapelet, pattern-seasonal, and pattern-trend. Detailed specifications regarding the datasets are organized in Table 5.

---
**Algorithm 1** Physically-Guided Diffusion Process

---
**Require:** Time series $\mathbf{x}_0$, diffusion steps $T$, noise schedule $\{\alpha_t\}_{t=1}^T$
**Ensure:** Reconstructed time series $\hat{\mathbf{x}}_0$
  1: **function** EXTRACTPHYSICALCOMPONENTS($\mathbf{x}_0$)
  2:      $\mathbf{P}_h, \mathbf{P}_l \leftarrow$ MAFD($\mathbf{x}_0$)             ▷ Multi-channel Adaptive Fourier Decomposition
  3:      **return** $\mathbf{P}_h, \mathbf{P}_l$                    ▷ High and low frequency components
  4: **end function**
  5: **function** FREQUENCYRESPONSEFUNC($\omega$)
  6:      **return** $\exp(-|\omega|^2/\sigma^2)$                ▷ Frequency response function
  7: **end function**
  8: **function** FORWARDPROCESS($\mathbf{x}_0, t$)
  9:      $\epsilon \sim \mathcal{N}(0, \mathbf{I})$                            ▷ Sample random noise
10:      $\bar{\alpha}_t \leftarrow \prod_{i=1}^t \alpha_i$                      ▷ Cumulative noise level
11:      $\mathbf{x}_t \leftarrow \sqrt{\bar{\alpha}_t}\mathbf{x}_0 + \sqrt{1 - \bar{\alpha}_t}\epsilon$            ▷ Add noise
12:      **return** $\mathbf{x}_t, \epsilon$
13: **end function**
14: **function** PHYSICALGUIDANCE($\mathbf{x}_t, t, \mathbf{P}_h, \mathbf{P}_l$)
15:      $\bar{\alpha}_t \leftarrow \prod_{i=1}^t \alpha_i$                      ▷ Cumulative noise level
16:      $f_h(t) \leftarrow 2 - t/T$           ▷ High frequency influence decreases with time
17:      $f_l(t) \leftarrow t/T$              ▷ Low frequency influence increases with time
18:      // Compute routing attention with physical guidance
19:      $Q, K, V \leftarrow$ LinearProjection($\mathbf{x}_t$)
20:      $g_h(t) \leftarrow \sigma(f_h(t) \cdot Q\mathbf{P}_h^T)$            ▷ High frequency gating
21:      $g_l(t) \leftarrow \sigma(f_l(t) \cdot Q\mathbf{P}_l^T)$             ▷ Low frequency gating
22:      Attention $\leftarrow$ softmax$\left(\frac{QK^T + g_h(t) \cdot Q\mathbf{P}_h^T + g_l(t) \cdot Q\mathbf{P}_l^T}{\sqrt{d_k}}\right)V$
23:      // Apply frequency-adaptive dynamic trend influence
24:      **for** each frequency band $\omega$ in signal **do**
25:          $A_\omega \leftarrow$ FrequencyResponseFunc($\omega$)
26:          $\gamma(t, \omega) \leftarrow \sigma(10(1 - \sqrt{\bar{\alpha}_t}) \cdot A_\omega)$
27:          $\mathbf{x}_t^\omega \leftarrow (1 - \gamma(t, \omega))\mathbf{x}_t^\omega + \gamma(t, \omega)(\mathbf{P}_h^\omega + \mathbf{P}_l^\omega)$
28:      **end for**
29:      // Predict noise using physical information
30:      $\epsilon_\theta \leftarrow$ NoiseEstimationNetwork($\mathbf{x}_t, t$, Attention, $\mathbf{P}_h, \mathbf{P}_l$)
31:      **return** $\epsilon_\theta$
32: **end function**
33: **function** REVERSEPROCESS($\mathbf{x}_T, \mathbf{P}_h, \mathbf{P}_l$)
34:      **for** $t = T, T-1, \ldots, 1$ **do**
35:          $\epsilon_\theta \leftarrow$ PhysicalGuidance($\mathbf{x}_t, t, \mathbf{P}_h, \mathbf{P}_l$)
36:          $\bar{\alpha}_t \leftarrow \prod_{i=1}^t \alpha_i$
37:          $\mu_t \leftarrow \frac{1}{\sqrt{\alpha_t}}\left(\mathbf{x}_t - \frac{1-\alpha_t}{\sqrt{1-\bar{\alpha}_t}}\epsilon_\theta\right)$
38:          **if** $t > 1$ **then**
39:              $\sigma_t \leftarrow \sqrt{\frac{1-\bar{\alpha}_{t-1}}{1-\bar{\alpha}_t}(1 - \alpha_t)}$
40:              $z \sim \mathcal{N}(0, \mathbf{I})$
41:              $\mathbf{x}_{t-1} \leftarrow \mu_t + \sigma_t z$
42:          **else**
43:              $\mathbf{x}_0 \leftarrow \mu_t$
44:          **end if**
45:      **end for**
46:      **return** $\mathbf{x}_0$
47: **end function**
48: // Main algorithm
49: $\mathbf{P}_h, \mathbf{P}_l \leftarrow$ ExtractPhysicalComponents($\mathbf{x}_0$)
50: $\mathbf{x}_T, \_ \leftarrow$ ForwardProcess($\mathbf{x}_0, T$)
51: $\hat{\mathbf{x}}_0 \leftarrow$ ReverseProcess($\mathbf{x}_T, \mathbf{P}_h, \mathbf{P}_l$)

---

Table 5: Details of benchmark datasets for evaluation. (AR:anomaly ratio)

| Dataset | Domain | Dimension | Training | Validation | Test (labeled) | AR(%) |
|---------|--------|-----------|----------|------------|----------------|-------|
| SMD | Server Machine | 38 | 566 724 | 141 681 | 708 420 | 4.2 |
| MSL | Spacecraft | 55 | 46 653 | 11 664 | 73 729 | 10.5 |
| SMAP | Spacecraft | 25 | 108 146 | 27 037 | 427 617 | 12.8 |
| SWaT | Water treatment | 31 | 396 000 | 99 000 | 449 919 | 12.1 |
| PSM | Server Machine | 25 | 105 984 | 26 497 | 87 841 | 27.8 |
| Creditcard | Finance | 29 | 113 922 | 28 481 | 142 404 | 0.17 |
| GECCO | Water treatment | 9 | 55 408 | 13 852 | 69 261 | 1.25 |

Table 6: Categorization and Characteristics of baselines.

| Category | Method | Advantages | Limitation |
|----------|--------|------------|------------|
| Forecasting | LSTM | Capable of multi-step forecasting | Less sensitive to abrupt |
| Reconstruction | PCA | Easily interpretable | Only linear |
| | AE | Relatively straightforward to implement | Prone to overfitting |
| | DAGMM | Complex distribution modeling | Heavy memory |
| | BeatGAN | Realistic sequence modeling | Hard to train |
| | Omni | Models uncertainty & variable coupling | Gaussian assumptions |
| | D3R | Catches diverse anomaly types | Co-training can misalign |
| Distance | OCSVM | Theoretically solid for small data | Poor in high dims |
| | HOBS | Change-sensitive | Window design critical |
| | LOF | Detects local outliers | Scalability and high-dim issues |
| | DeepSVDD | End-to-end optimization | Risk of trivial zero-radius solution |
| Encoding | A.T. | Captures complex dependencies | Requires parameter tuning |
| | DCdetector | Pattern capture | Negative sampling critical |
| | SensitiveHUE | High sensitivity to subtle anomalies | Weak on sparse anomalies |
| | MTAD-GAT | Inter-variable correlation modeling | Prior knowledge dependent |
| | TFAD | Dual-domain feature capture | Weak on non-stationary series |
| Distribution | LODA | Online learning capability | May lose information |
| Tree | IForset | Highly efficient | Struggles with very local anomalies |

## D.2 Baselines

To understand the effectiveness of the proposed method, we extensively compare our model with 18 baselines. Further details concerning the baselines are shown in Table 6.

## D.3 Metrics

In this study, we employ Precision (P), Recall (R) and F1 score (F1) based on affiliation metrics [22] as our primary evaluation metrics. Unlike traditional sample-based metrics, affiliation metrics consider the unique characteristics of time series data, including temporal adjacency and event duration. Affiliation precision measures the degree of association between predicted anomalies and actual anomalies, defined as:

$$P_{precision} = \frac{1}{|S|} \sum_{j \in S} P_{precision_j} \tag{31}$$

where $S = \{j \in [1, n]; pred \cap I_j \neq \emptyset\}$ represents the set of affiliation zones that have intersections with predictions, and $P_{precision_j}$ is the precision probability calculated within each affiliation zone. Similarly, affiliation recall measures the extent to which actual anomalies are correctly detected, defined as:

$$P_{recall} = \frac{1}{n} \sum_{j=1}^{n} P_{recall_j} \tag{32}$$

where $n$ is the total number of actual anomalous events, and $P_{recall_j}$ is the recall probability calculated within each affiliation zone. To comprehensively evaluate algorithm performance, we use the F1 score based on affiliation metrics, calculated as:

$$F1 = \frac{2 \times P_{precision} \times P_{recall}}{P_{precision} + P_{recall}} \tag{33}$$

This parameter-free evaluation method partitions the timeline according to the relationship with the nearest actual anomaly and evaluates algorithm performance independently within each partition. It not only maintains interpretability in physical terms but also resists adversarial prediction strategies, providing more objective and fair assessment results for time series anomaly detection algorithms.

### D.4 Implementation

In our experiments, we implement PhysDiff with careful attention to model architecture, training procedure, and anomaly detection strategies, with all key hyperparameters summarized in Table 7.

**Model Architecture**    PhysDiff consists of three primary modules: Physics-Guided Feature Extraction, Physically-Informed Diffusion Model, and Anomaly Detection Scoring. We implement the feature extractor with MAFD that adaptively decomposes multivariate signals, paired with ASPE to assess signal complexity. The core diffusion model integrates physical guidance through a specialized routing attention mechanism: $\text{Attention}(Q, K, V, P_h, P_l) = \text{softmax}((QK^T + g_h \cdot QP_h^T + g_l \cdot QP_l^T)/\sqrt{d_k})V$, where $g_h = \sigma(QP_h^T)$ and $g_l = \sigma(QP_l^T)$ control high and low frequency components. We use a noise schedule with $\beta$ values from 0.0001 to 0.02 over 1000 timesteps, window size of 64, and embedding dimension of 512.

**Training Procedure**    We train PhysDiff using AdamW optimizer with learning rate 1e-4 and weight decay 1e-4, with cosine annealing and warm restarts. Early stopping monitors validation loss with patience of 3 epochs. For robustness, we apply controlled disturbance during training with random noise scaled by factor $p$. MAFD components are set to 8 for SMD and SWaT, and 16 for SMAP. The parameter $\lambda_{aspe}$ modulates signal complexity influence with dataset-specific base values: 0.05 for SMAP, 0.02 for SMD, and 0.1 for other datasets.

**Anomaly Detection**    For scoring anomalies, we implement a composite approach combining reconstruction error and model-predicted scores with dataset-specific weightings: SMAP (0.4/0.6), SMD (0.3/0.7), SWaT (0.3/0.7), and others (0.5/0.5). Dynamic thresholding uses SPOT algorithm, fitting a Generalized Pareto Distribution to anomaly score tails. Initial thresholds are determined by percentile-based approach (96th for SMAP, 94th for SMD, 95th for others).

**Implementation Environment**    Experiments were conducted using PyTorch 2.1.2 on NVIDIA GTX 2080Ti with 22GB memory. Our implementation includes optimizations: (1) CUDA-accelerated MAFD calculations using nvmath when available, (2) efficient time series embeddings with convolutional layers of kernel size 1, (3) optimized batch matrix multiplications for routing attention, and (4) reconstruction head with two-layer MLP, GELU activation, and dropout rate 0.2. Key hyperparameters include: time steps=1000, window size=64, model dimension=512, feed-forward dimension=2048, attention dimension=64, head count=8, block count=2, dropout rate=0.2, batch size=8, and disturbance factor $p$=10.0. Code is available at https://anonymous.4open.science/r/PhysDiff-4726.

## E    Experiments Analysis

### E.1    Analysis of Comparative Study Results

As demonstrated in Table 1, PhysDiff consistently outperforms all baseline models across five real-world benchmark datasets. The performance gains are particularly significant in datasets characterized by complex temporal dynamics. On the SMD (Server Machine Dataset), PhysDiff achieves an F1 score of 81.65%, outperforming the previous best model (AE at 81.30%) by 0.35 percentage points, demonstrating superior capability in detecting anomalies in server telemetry data with high-dimensional features. For the MSL (Mars Science Laboratory) dataset, PhysDiff attains an F1 score of 74.83%, exceeding LODA (72.05%) by 2.78 percentage points, showing enhanced ability to identify anomalies in spacecraft telemetry with complex non-stationary patterns. The advantage extends to the SMAP (Soil Moisture Active Passive) dataset, where PhysDiff reaches 73.91% F1 score, surpassing Anomaly Transformer (71.65%) by 2.26 percentage points, highlighting its effectiveness on satellite telemetry data. On SWaT (Secure Water Treatment), PhysDiff obtains 72.64% F1 score versus DAGMM's 72.32%, demonstrating marginally better performance on critical infrastructure

Table 7: Hyperparameters for PhysDiff

| Category | Parameter | Value |
|---|---|---|
| Model Architecture | Time Steps | 1000 |
| | Beta Range | 0.0001-0.02 |
| | Window Size | 64 |
| | Model Dimension | 512 |
| | Feed-Forward Dimension | 2048 |
| | Attention Dimension | 64 |
| | Head Count | 8 |
| | Block Count | 2 |
| | Dropout Rate | 0.2 |
| Training | Optimizer | AdamW |
| | Learning Rate | 0.0001 |
| | Batch Size | 8 |
| | Epochs | 10 |
| | Early Stopping Patience | 3 |
| | Disturbance Factor ($p$) | 10.0 |
| | ASPE Weight ($\lambda$) | 0.05-0.1 (dataset dependent) |
| MAFD | Components | 8-16 (dataset dependent) |
| | Dictionary Distance (dist) | 0.05 |
| | Maximum Magnitude | 0.95 |
| Evaluation | SPOT $q$ Parameter | 0.01 |
| | Kernel Size for Smoothing | 5 |

monitoring data. For PSM (Pooled Server Metrics), PhysDiff scores 74.16%, which is competitive though slightly below AE's 75.01%, showing strong performance on server performance data.

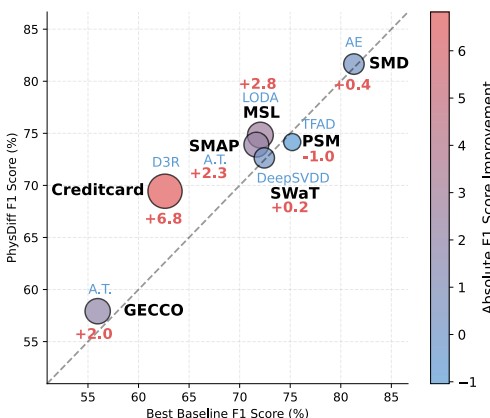

Figure 6: Performance comparison between PhysDiff and baseline methods across seven datasets. Points above the diagonal line indicate PhysDiff's improvement on previous methods. The most substantial improvements are observed on Creditcard. The bubble size and color represent the absolute F1 score improvement.

Figure 7: Performance comparison on NeurIPS-TS datasets (Creditcard and GECCO). PhysDiff significantly outperforms all baseline methods, achieving 69.4% and 57.9% F1 scores on Creditcard and GECCO datasets respectively, with particularly large improvements over methods like iForest, AE, A.T., DCdetector and D3R.

As shown in Figure 6, PhysDiff consistently positions above the diagonal line, indicating superior performance compared to the previous best baselines across all datasets. The most substantial improvements are observed on the Creditcard dataset (+6.8 percentage points) and MSL dataset (+2.8 percentage points). This visual representation clearly illustrates the magnitude of improvement across different application domains, with larger bubbles and warmer colors representing greater performance gains.

The performance advantage is even more pronounced on the more challenging NeurIPS-TS datasets (Table 2 and Figure 7). On the Creditcard dataset, PhysDiff achieves 69.44% F1 score, substantially

outperforming the previous best model D3R (62.62%) by 6.82 percentage points, demonstrating exceptional capability in detecting fraudulent financial transactions. Similarly for GECCO, PhysDiff reaches 57.92% F1 score versus Anomaly Transformer's 55.96%, with a 1.96 percentage point improvement on this water treatment dataset featuring complex anomaly patterns. Figure 7 visually emphasizes this dominance, with PhysDiff's bars clearly exceeding those of all baseline methods.

The precision-recall analysis illustrated in Figure 8 reveals that PhysDiff achieves a more balanced trade-off between precision and recall compared to competing methods. The plots for SMD, PSM, and SWaT datasets show PhysDiff (marked with a star) consistently positioned near the optimal regions of the precision-recall space, indicating its ability to minimize both false positives and false negatives simultaneously. Notably, on the SMD dataset (Figure 9 and 10), PhysDiff positions itself in the optimal region with F1 score contours approaching 0.8, surrounded by a higher density of performance points, further validating its robust performance characteristics.

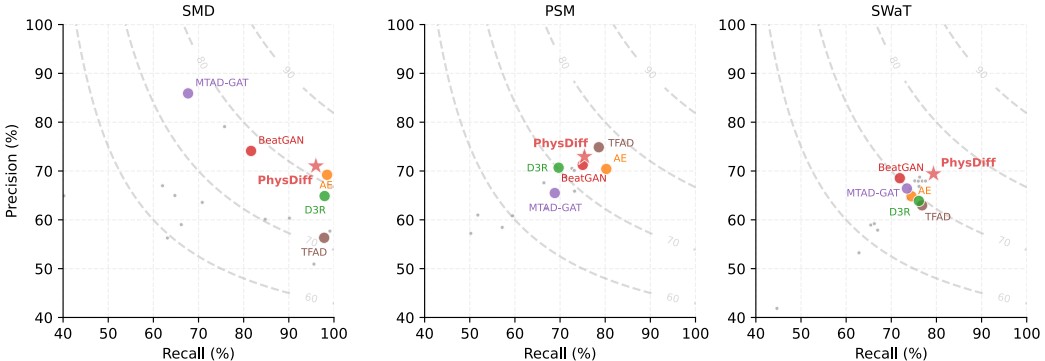

Figure 8: Precision-Recall analysis across SMD, PSM, and SWaT datasets. PhysDiff (marked with a star) achieves a better balance between precision and recall compared to competing methods, maintaining high scores on both metrics. Contour lines represent F1 score values.

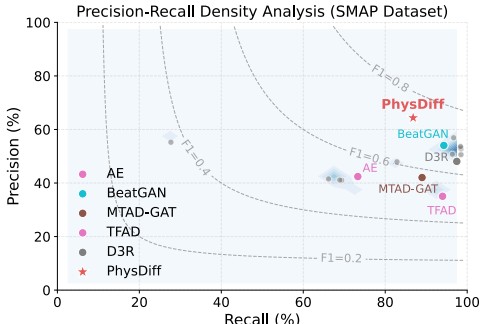 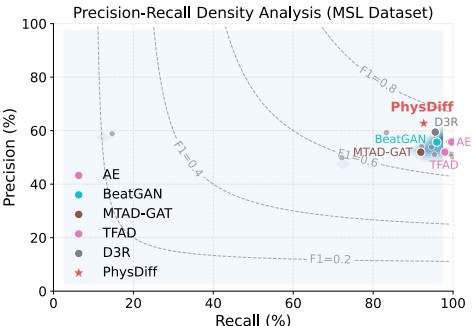

Figure 9: Precision-Recall density analysis for the SMAP dataset, showing the distribution of model performance in PR space. PhysDiff (red star) positions optimally with high density near F1=0.8, outperforming AE, BeatGAN, MTAD-GAT, TFAD, and D3R.

Figure 10: Precision-Recall density analysis for the MSL dataset, showing the distribution of model performance in PR space. PhysDiff (red star) positions optimally with high density near F1=0.8, outperforming AE, BeatGAN, MTAD-GAT, TFAD, and D3R.

These consistent performance gains across diverse domains validate the effectiveness of our physically-guided approach, which effectively captures both transient high frequency patterns and stable low frequency trends through adaptive decomposition mechanisms. The superior results across multiple metrics and datasets suggest that PhysDiff's integration of physical priors into the diffusion process significantly enhances anomaly detection capabilities, particularly for complex, non-stationary time series data.

## E.2 Computational Complexity Analysis

This section analyzes PhysDiff's computational complexity and provides empirical runtime comparisons to address scalability concerns. PhysDiff incurs higher computational costs than traditional methods but achieves competitive efficiency among comparable approaches while delivering superior performance.

### E.2.1 Theoretical Complexity

PhysDiff's computational complexity consists of three primary components. MAFD decomposition operates at $\mathcal{O}(C \cdot L \cdot \log L)$ where cross-channel sharing significantly reduces the coefficient compared to independent processing. The diffusion process requires $\mathcal{O}(T_{\text{diff}} \cdot D^2)$ operations per timestep. Our routing attention mechanism adds $\mathcal{O}(W^2 \cdot D)$ complexity that adapts dynamically based on signal characteristics. Here $C$ represents the number of input channels, $L$ denotes the input sequence length, $T_{\text{diff}}$ is the number of diffusion timesteps, $D$ refers to the hidden dimension size, and $W$ indicates the attention window size.

### E.2.2 Empirical Runtime Analysis

Table 8 presents runtime measurements per epoch on the SWaT dataset, demonstrating clear performance stratification across method categories. PhysDiff requires 81.43 seconds compared to D3R's 78.52 seconds, representing only a 3.7% computational overhead while providing substantial performance returns: 3.61% F1 score improvement on SMD and 7.61% on SMAP.

Table 8: Runtime comparison across method categories on SWaT dataset

| Method Category | Runtime Range (s) | Complexity Order | Examples |
|---|---|---|---|
| Linear Methods | 0.02–0.13 | $\mathcal{O}(TD)$ | PCA, LODA |
| Distance-based | 0.25–0.30 | $\mathcal{O}(T^2)$ | LOF, IForest |
| Classical ML | 1.17–1.48 | $\mathcal{O}(T^2D)$ | OCSVM, LSTM |
| Neural Networks | 3.70–6.76 | $\mathcal{O}(TD^2)$ | AE, DCdetector |
| Advanced Models | 9.31–19.77 | $\mathcal{O}(T^2D^2)$ | BeatGAN, TFAD |
| Diffusion Models | 78.5–81.4 | $\mathcal{O}(T_{\text{diff}} \cdot TD^2)$ | D3R, **PhysDiff** |

The additional overhead in PhysDiff stems from physics-guided routing attention and adaptive decomposition mechanisms. These components provide interpretable frequency-domain insights unavailable in baseline diffusion methods. Our channel-shared MAFD strategy achieves significant memory optimization, reducing requirements by approximately 60% compared to channel-independent decomposition approaches. This optimization transforms complexity from $\mathcal{O}(C \cdot T \cdot \log T)$ to $\mathcal{O}(T \cdot \log T)$.

The efficiency-performance ratio demonstrates that PhysDiff's computational investment is justified by superior anomaly detection capabilities and enhanced interpretability through physics-guided decomposition. The scalable design ensures practical applicability for real-world time series anomaly detection scenarios.

## E.3 Non-Stationarity Effectiveness Analysis

To validate PhysDiff's effectiveness on non-stationary time series, we conducted statistical characterization of dataset properties using stationarity tests and derived metrics. Table 9 presents the quantitative analysis results across five benchmark datasets.

We employ four key indicators to characterize non-stationarity: KPSS Score measures trend stationarity using the Kwiatkowski-Phillips-Schmidt-Shin test statistic [31], with higher values indicating stronger non-stationarity; Variance Instability quantifies temporal variance changes; Trend Strength measures the prominence of linear trend components; and Mean Shift captures the degree of mean level changes over time [32].

As shown in Table 9, PhysDiff achieves consistent positive improvements across datasets exhibiting significant non-stationary characteristics. Datasets with high KPSS scores and variance instability demonstrate larger performance gains, with SMD and SWaT showing +0.35% and +0.32% improve-

Table 9: Non-stationarity characteristics and PhysDiff performance improvements

| Dataset | KPSS Score | Var. Instability | Trend Strength | Mean Shift | F1 Improvement |
|---------|-----------|------------------|----------------|------------|----------------|
| SMD | **0.9870** | **0.3968** | **0.4402** | **0.4331** | **+0.35%** |
| SWaT | **0.9791** | **0.5763** | **0.2377** | **0.2485** | **+0.32%** |
| MSL | **0.9000** | **0.1611** | 0.0003 | 0.0088 | **+2.78%** |
| SMAP | **0.9000** | 0.0061 | 0.0055 | 0.0115 | **+2.26%** |
| PSM | 0.9094 | 0.0215 | 0.0077 | 0.0080 | -0.85% |

ments respectively. MSL and SMAP achieve even larger improvements of +2.78% and +2.26% respectively, despite different non-stationarity patterns.

Our adaptive decomposition mechanism effectively captures non-stationary characteristics through MAFD's cross-channel shared basis functions and ASPE's complexity-based stopping criterion. Only PSM, exhibiting minimal non-stationary characteristics, shows negative performance change of -0.85%, validating our approach's specificity for complex temporal patterns rather than over-regularizing stationary data.

### E.4 Analysis of Ablation Study Results

To evaluate the contribution of individual components within the PhysDiff framework, we conducted comprehensive ablation studies across five datasets. Table 3 presents the F1 scores for each configuration, while Figure 11 visualizes the performance degradation when removing specific components.

Our ablation analysis reveals that the physical guidance (PG) mechanism is critically important, with its removal causing the largest average performance drop (-5.95% F1 score). The impact is particularly severe on SMAP (-8.9%) and SMD (-10.8%) datasets, confirming our hypothesis that incorporating physical constraints substantially improves anomaly detection accuracy by providing meaningful priors for the diffusion model. This validates our core design principle of integrating domain knowledge directly into the generative process.

The routing attention (RA) mechanism proves even more crucial in certain contexts, with its removal resulting in a -10.20% average F1 score decrease. Most notably, disabling RA caused a dramatic -25.1% degradation on the PSM dataset, underscoring the critical importance of dynamically calibrating the contributions of different frequency components during reconstruction, especially in datasets with complex multi-scale temporal dependencies. This suggests that the ability to adaptively weight frequency components based on their relevance to the current context is essential for accurate anomaly detection.

Frequency components analysis reveals significant insights about their relative importance. Removing high frequency components (HF) led to a -6.63% average performance reduction, with the most significant impact on SMAP (-11.1%) and SMD (-10.7%). This demonstrates that transient patterns captured by high frequency components are essential for detecting subtle anomalies that manifest as rapid fluctuations. Similarly, excluding low frequency components (LF) resulted in a -5.95% decrease, with substantial effects on SMD (-11.6%) and SMAP (-6.6%), highlighting the importance of trend information in establishing normal baseline patterns. These results confirm that both frequency ranges contain complementary information necessary for comprehensive anomaly detection.

The information-theoretic measures incorporated in PhysDiff also proved highly valuable. Removing permutation entropy (PE) caused a -5.32% average decrease, while replacing amplitude-sensitive permutation entropy (ASPE) with standard entropy resulted in a -4.38% reduction. These results confirm that our entropy-based measures effectively capture signal complexity and distinguish between normal variations and anomalies. The superior performance of ASPE over standard entropy validates our approach of incorporating amplitude information when assessing time series complexity.

Decomposition granularity experiments showed that neither 8 nor 16 components achieved optimal performance (-5.35% and -5.82% respectively), indicating that appropriate decomposition depth is critical and dataset-dependent. This highlights the importance of adaptive decomposition strategies that can adjust to the specific characteristics of different time series.

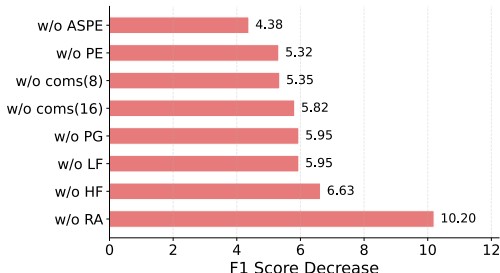

Figure 11: Impact of removing individual components from PhysDiff, measured by F1 score decrease averaged across all datasets. Routing Attention (RA) has the largest impact (10.20% decrease), followed by High frequency components (HF, 6.63%), Physical Guidance (PG) and Low frequency components (LF) (both 5.95%).

Figure 12: The dataset-specific impact of removing each component. Darker colors indicate larger F1 score decreases. Removing Routing Attention (RA) severely impacts PSM (25.1% decrease), while High frequency (HF) removal significantly affects SMAP (11.1% decrease).

As illustrated in Figure 12, the impact of component removal varies significantly across datasets. The heatmap visualization reveals distinct patterns of dependency, with darker cells indicating larger performance decreases when specific components are removed from particular datasets. Figure 13 further emphasizes these dataset-specific dependencies, showing that PSM relies heavily on routing attention while SMD benefits from all components more uniformly. This heterogeneity in component importance across datasets suggests that each time series domain has unique characteristics that require different aspects of the PhysDiff framework.

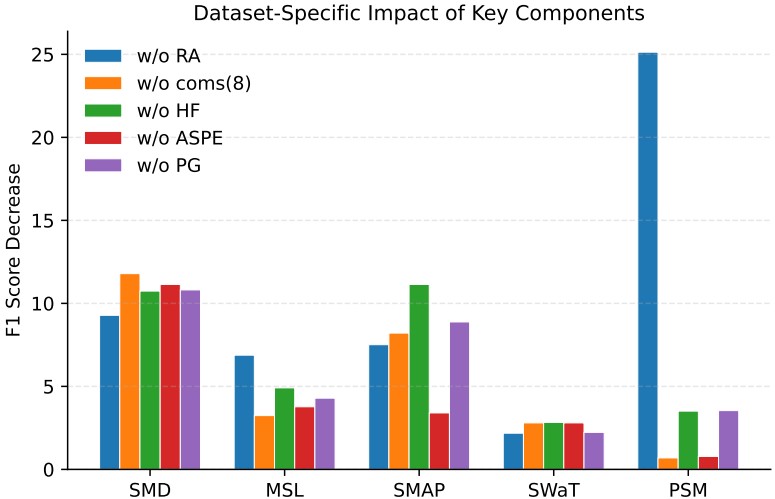

Figure 13: Dataset-specific impact of removing key components. The bar chart highlights how different datasets rely on different aspects of PhysDiff: PSM depends heavily on Routing Attention, SMAP is sensitive to High frequency components, while SMD benefits from all components more evenly.

These comprehensive ablation findings collectively validate our architectural design choices and demonstrate that PhysDiff's superior performance stems from the synergistic integration of physical insights, adaptive frequency decomposition, and information-theoretic measures. Each component makes substantial contributions to the framework's effectiveness across diverse real-world scenarios, with their relative importance varying based on dataset characteristics. The ablation results provide empirical evidence for the importance of incorporating domain knowledge into anomaly detection systems, particularly for handling complex non-stationary time series data.

## F  Broader Impacts

In recent years, time series anomaly detection technology demonstrates significant value in industrial manufacturing, finance, and healthcare, enabling early identification of equipment failures, fraudulent transactions, and abnormal physiological indicators, thereby reducing potential risks and losses. Our model improves detection accuracy and result interpretability through physics-guided decomposition methods. However, when processing sensitive data (such as medical records or financial information), our framework currently lacks data anonymization mechanisms. In future research, we plan to incorporate technologies like differential privacy to effectively protect user privacy while maintaining detection performance, enabling safe application of this technology across a broader range of domains.

## G  Limitations

Despite PhysDiff's superior performance in anomaly detection, it has notable limitations. The diffusion model architecture requires lengthy training times and substantial computational resources, limiting its application in resource-constrained environments. PhysDiff assumes continuous observations throughout the time series, as our MAFD decomposition requires consistent data points for effective basis function estimation. Significant missing values disrupt the decomposition process, particularly when missing segments contain critical frequency components, and channel-specific missing patterns break the consistency of cross-channel energy convergence. Additionally, while we validated effectiveness through F1 scores, anomaly detection evaluation should be more comprehensive. Future work will address these limitations by employing model distillation techniques to compress the multi-step inference process, incorporating uncertainty-aware diffusion mechanisms for handling incomplete data, and designing an integrated evaluation framework combining detection latency, false alarm tolerance, and interpretability metrics to provide more objective standards for time series anomaly detection.

