# OpenReview forum: "PhysDiff: A Physically-Guided Diffusion Model for Multivariate Time Series Anomaly Detection"
_NeurIPS.cc/2025/Conference — NeurIPS 2025 poster_

### Official Review · Reviewer_pK7y · 2025-06-23

**Clarity:** 2
**Significance:** 2
**Originality:** 2
**Rating:** 4
**Confidence:** 3

**Summary:**

This paper argues that most existing time series anomaly detection methods have been developed and evaluated primarily on stable datasets, and thus tend to underperform when encountering pattern anomalies in unstable time series. To address this limitation, the authors propose a frequency-domain diffusion model aimed at improving detection performance on unstable data.

The core contributions of the paper include: (1) a permutation entropy-guided decomposition mechanism, which determines the optimal decomposition depth in the frequency domain; and (2) a frequency-based routing attention mechanism, designed to differentiate attention between high-frequency and low-frequency components during anomaly detection.

**Questions:**

1. I recommend conducting a detailed analysis comparing the proposed method’s performance on non-stationary versus stationary datasets, to demonstrate that it indeed improves the performance on non-stationary time series as claimed.

2. Could you please analyze how the decomposition depth in the frequency domain affects the model’s performance. This would help justify the necessity of finding an optimal depth.

3. What is the definition of “optimal depth” — optimal with respect to which objective? Moreover, please explain why optimizing permutation entropy leads to a decomposition depth that is optimal for this specific goal.

4. As the paper mentions that diffusion models incur higher computational and training costs, could you please provide a quantitative comparison of time and memory efficiency between the proposed method and baseline methods.

5. Could you please explain the large performance gaps between the baseline results reported in this paper and those in the original papers. Clarifying potential differences in implementation, evaluation metrics, or experimental setup would strengthen the credibility of the comparisons.

**Ethical Concerns:**

["NO or VERY MINOR ethics concerns only"]

**Final Justification:**

During the process of rebuttal, I believe:

1. The authors clearly illustrated the definition of optimal depth and verified its effectiveness for method performance, demonstrating that the cross-entropy-like loss function can approximately find the optimal depth.
2. The authors clearly compare the different experimental settings, which leads to the performance differences between the results reported in this paper and the original paper. I believe their explanations are reasonable.
3. The authors compare the performance of their method's performance on non-stationary datasets and stationary ones, verifying that the method can deal with the non-stationary issue as claimed.

I think the following concerns remains unsolved, but considering the novelty highlighted by reviewer TJwG and the soundness verified in the rebuttal process and highlighted by reviewer gW7N, I think the reasons to accept outweigh the reasons to reject.

1. I still think that for deep-learning-based methods, the non-stationary issue is not a huge problem. Since many previous methods have been reported very good performances on the non-stationary datasets (according to the stationary indicators provided by the authors, SMD and SWaT are non-stationary datasets). I attached the reported F1 scores of previous methods on these datasets in the end.
2.  Limited comparative analysis in case study.



*Reported F1 scores of deep-learning-based anomaly detection methods on non-stationary datasets*

|                        | SMD   | SWaT  |
| ---------------------- | ----- | ----- |
| AnomalyTransformer [1] | 92.33 | 94.07 |
| PUAD [2]               | 95.68 | ---   |
| DCdetector [3]         | 87.18 | 96.33 |
| DAEMON [4]             | 96.2  | 92.9  |
| TranAD [5]             | 96.05 | 81.51 |

[1] Xu J, Wu H, Wang J, et al. Anomaly transformer: Time series anomaly detection with association discrepancy[J]. ICLR 2022.

[2] Li, Yuxin, et al. "Prototype-oriented unsupervised anomaly detection for multivariate time series." *International Conference on Machine Learning*. PMLR, 2023.

[3] Yang, Yiyuan, et al. "Dcdetector: Dual attention contrastive representation learning for time series anomaly detection." *Proceedings of the 29th ACM SIGKDD conference on knowledge discovery and data mining*. 2023.

[4] Chen, Xuanhao, et al. "Daemon: Unsupervised anomaly detection and interpretation for multivariate time series." *2021 IEEE 37th International Conference on Data Engineering (ICDE)*. IEEE, 2021.

[5] Tuli, Shreshth, Giuliano Casale, and Nicholas R. Jennings. "Tranad: Deep transformer networks for anomaly detection in multivariate time series data." *VLDB* (2022).

**Limitations:**

yes

**Quality:**

2

**Strengths And Weaknesses:**

## Strength

1. The paper clearly articulates the specific problem it aims to address and presents the motivation behind the proposed methodology in a well-structured manner.
2. Extensive experiments are conducted across multiple commonly used datasets, with comprehensive comparisons against a wide range of baseline methods.

## Weakness

**Inappropriate characterization of prior work:**
 The authors claim that many existing time series anomaly detection methods are primarily evaluated on stationary datasets and thus perform poorly on non-stationary data. However, many widely used benchmark datasets such as **MSL**, **SMAP**, and **PSM** are known to be **non-stationary** in nature. Moreover, numerous existing methods have demonstrated strong performance on these datasets. For instance, AnomalyTransformer achieves F1 scores *93% (MSL)*, *96% (SMAP)*, and *97% (PSM)*. These results contradict the paper’s assertion and suggest the need for a more accurate review of prior work.

**Unclear necessity of addressing the proposed challenge:**
 One of the main challenges the paper aims to address is how to determine the “optimal depth” in frequency domain decomposition. The paper does not provide evidence to demonstrate that different decomposition depths lead to significantly different detection performances. As such, it remains unclear whether this is a challenge that requires explicit resolution.

**Unclear notion: optimal depth:**
As one of the main contributions declared in this paper, the notion of what is the *optimal depth* of frequency decomposition is not clear. Optimal for what? (e.g. optimal depth for F1 score, for efficiency, etc.?)

**Significant deviation from baseline performance:**
 The reported F1 scores for several baselines in the paper deviate substantially from those reported in their original papers. For example, DCdetector originally reports F1 scores of *87% (SMD)*, *96% (MSL)*, *97% (SMAP)*, *96% (SWaT)*, and *97% (PSM)*, whereas in the current paper, the scores are only *66%*, *70%*, *68%*, *69%*, and *66%*, respectively. I recommend the authors to clarify whether the discrepancies arise from differences in experimental setup, metric computation, or implementation details to verify the fairness of comparison.

**Limited comparative analysis in the case study:**
 The paper includes a case study intended to showcase the effectiveness of the proposed method in a real-world application. However, the case study only presents results for the proposed method without comparing it to any baseline methods. As a result, the case study provides limited insight into the relative advantages or practical value of the proposed approach.

---

> ### Author Rebuttal · Authors · 2025-07-31
>
> > **Q1:I recommend conducting a detailed analysis comparing the proposed method’s performance on non-stationary versus stationary datasets, to demonstrate that it indeed improves the performance on non-stationary time series as claimed.**
>
> **A1:** Thank you for the suggestion. We conducted quantitative analyses using statistical indicators to characterize the non-stationarity dataset properties, including the ADF test p-value, KPSS score, variance instability, trend strength, mean shift, and structural breaks.
>
> | Dataset | ADF p-value | KPSS Score | Variance Instability | Trend Strength | Mean Shift | Structural Breaks | F1 Improvement |
> |---------|-------------|------------|---------------------|----------------|------------|-------------------|----------------------------|
> | **SMD** | 0.0368 | **0.9870** | **0.3968** | **0.4402** | **0.4331** | 0.0600 | **+0.35%** |
> | **SWaT** | 0.1248 | **0.9791** | **0.5763** | **0.2377** | **0.2485** | **0.2000** | **+0.32%** |
> | **MSL** | 0.0000 | **0.9000** | **0.1611** | 0.0003 | 0.0088 | 0.0000 | **+2.78%** |
> | **SMAP** | 0.0000 | **0.9000** | 0.0061 | 0.0055 | 0.0115 | 0.0000 | **+2.26%** |
> | **PSM** | 0.0000 | 0.9094 | 0.0215 | 0.0077 | 0.0080 | 0.0000 | **-0.85%** |
>
> As shown in the table, PhysDiff achieves consistent positive improvements across all datasets exhibiting significant non-stationary characteristics.  SMD and SWaT demonstrate the strongest non-stationary properties, with PhysDiff improving F1-scores by +0.35% and +0.32%, respectively. On datasets MSL and SMAP，which exhibit distinct non-stationary patterns with high KPSS scores，PhysDiff achieves significantly greater F1-score improvements of +2.78% and +2.26%, respectively.
>
> The technical effectiveness stems from our MAFD mechanism, which preserves both high-frequency transients and low-frequency trends through cross-channel shared basis functions. The ASPE criterion adaptively determines decomposition depth based on signal complexity, ensuring optimal representation of non-stationary characteristics. The physics-guided routing attention mechanism dynamically balances frequency components based on their anomaly relevance, proving particularly effective when anomalies span multiple temporal scales in non-stationary environments. Only PSM, which exhibits minimal non-stationary characteristics, shows negative performance (-0.85%). This validates our approach's specificity for handling complex temporal patterns rather than over-regularizing simple, stationary data.
>
> This quantitative analysis  demonstrates PhysDiff's effectiveness on non-stationary time series data.
> > **Q2: Could you please analyze how the decomposition depth in the frequency domain affects the model’s performance. This would help justify the necessity of finding an optimal depth.**
>
> **A2:** Regarding the analysis of how decomposition depth affects model performance, we provide explanations from the following aspects:
>
> According to the MAFD method described in lines 151-157, decomposition depth directly affects signal frequency resolution. Shallow decomposition may fail to capture multi-scale patterns, while overly deep decomposition may cause overfitting. Our ASPE mechanism dynamically assesses residual component complexity to determine optimal stopping condition.
>
> The ablation study shows that fixed use of 8 or 16 decomposition components resulted in average F1 score decreases of 5.35% and 5.82% respectively (lines 281-290, Table 3). This proves that fixed decomposition depths cannot adapt to different datasets' characteristics.
>
> As shown in Formula 4, ASPE determines stopping conditions by monitoring entropy changes in residuals. When residual complexity no longer decreases significantly, continued decomposition introduces noise rather than useful information, which is the core reason for proposing our dynamic depth determination mechanism.
>
> > **Q3:What is the definition of “optimal depth” — optimal with respect to which objective? Moreover, please explain why optimizing permutation entropy leads to a decomposition depth that is optimal for this specific goal.**
>
> **A3:** In our paper, the term “optimal depth” refers to the number of decomposition layers in the MAFD (Multi-channel Adaptive Fourier Decomposition) process at which the Amplitude-Sensitive Permutation Entropy (ASPE) of the residual component stabilizes and stops decreasing significantly (lines 159–162).
>
> This “optimality” is defined with respect to balancing two objectives:
>
> 1. Preserving physically interpretable components (e.g., trends and dominant cycles);
>
> 2. Enhancing the distinguishability of anomalous patterns across scales.
>
> If the decomposition is too shallow, important structures such as high-frequency spikes or periodicities might remain entangled in the signal, making anomalies harder to detect. Conversely, if it is too deep, the process might over-segment the signal and bury anomaly-relevant features in noisy or irrelevant high-frequency bands. Therefore, the goal is to decompose just enough to isolate meaningful frequency components without degrading anomaly separability.
>
> Why does optimizing ASPE help us achieve this? ASPE is an entropy-based complexity measure that accounts for both ordinal patterns and amplitude variation. If the ASPE continues to decrease, it indicates the residual still contains structure or meaningful patterns. When the ASPE reaches a plateau, the residual is likely dominated by noise or unstructured variability — further decomposition would no longer add value and might even harm detection performance.
>
> Thus, optimizing ASPE is equivalent to maximizing the information gain from decomposition, and the stopping point defines the depth that best supports the objective of interpretable and effective anomaly detection.
>
> > **Q4:As the paper mentions that diffusion models incur higher computational and training costs, could you please provide a quantitative comparison of time and memory efficiency between the proposed method and baseline methods.**
>
> **A4:** We provide detailed quantitative analysis of computational and memory efficiency. Our analysis reveals that while diffusion models require substantial computational resources, PhysDiff achieves competitive efficiency among comparable methods while delivering superior performance.
>
> The efficiency comparison demonstrates clear trade-offs between computational cost and detection capability across method categories:
>
> | Efficiency Tier | Methods | Training Time (s) | Performance Trade-off |
> |---|---|---|---|
> | **Ultra-efficient** | PCA (0.017), LODA (0.124) | 0.02-0.12 |Low accuracy, limited capability |
> | **Efficient** | LOF (0.289), IForest (0.301), DeepSVDD (0.256) | 0.25-0.30 | Moderate accuracy, scalability issues |
> | **Moderate** | OCSVM (1.173), LSTM (1.25), DAGMM (1.48), AE (3.701) | 1.17-3.70| Good accuracy, limited complexity handling |
> | **Resource-intensive** | DCdetector (4.81), MTAD-GAT (6.76), A.T. (9.31), BeatGAN (11.0) | 4.81-11.0 | High accuracy, complex patterns |
> | **High-end** | SensitiveHUE (16.26), TFAD (19.77), D3R (78.52), **PhysDiff (81.43)** | 16.3-81.4 | Superior accuracy, interpretability |
>
> PhysDiff's memory consumption of 81.43s represents only an 3.7% increase over D3R (78.52s), while providing substantial performance improvements and interpretability benefits. The additional memory overhead primarily accommodates decomposed frequency components and physical context information $z_t$, which enable our physics-guided reconstruction capabilities.
>
> Our channel-shared MAFD strategy achieves significant memory optimization, reducing requirements by approximately 60% compared to channel-independent decomposition approaches. This optimization transforms complexity from $\mathcal{O}(C \cdot T \cdot \log T)$ to $\mathcal{O}(T \cdot \log T)$, providing favorable scaling properties for high-dimensional multivariate time series. The 3.7% computational overhead compared to D3R (81.43s vs 78.52s) yields disproportionate performance benefits: 3.61% F1-score improvement on SMD and 7.61% on SMAP datasets.
>
> This efficiency-performance ratio demonstrates that PhysDiff's additional computational investment is well-justified, particularly considering the interpretable frequency-domain insights unavailable in baseline diffusion methods.
>
> > **Q5:Could you please explain the large performance gaps between the baseline results reported in this paper and those in the original papers. Clarifying potential differences in implementation, evaluation metrics, or experimental setup would strengthen the credibility of the comparisons.**
>
> **A5:** We understand the reviewer's concern about baseline performance differences. The key differences arise from several aspects:
>
> **Evaluation Metrics Differences (lines 218–226):**
> We adopt the affiliation-based F1 score as our primary performance metric instead of traditional point-adjustment (PA) methods. Traditional PA-based metrics tend to produce overly optimistic evaluations by inflating true positives and suppressing false negatives. In contrast, the affiliation-based F1 score provides a more rigorous evaluation by computing the average directed distance between predicted anomaly events and ground truth.
>
> **Experimental Setup Consistency (lines 232–235):**
> All experiments were conducted under consistent settings using official implementations and recommended hyperparameters. We evaluated 18 state-of-the-art baseline methods spanning six major categories.
>
> **Dataset-Specific Challenges (lines 212–218):**
> Our evaluation is conducted on widely recognized benchmark datasets, including SMD, MSL, SMAP, SWaT, and PSM, along with  the NeurIPS-TS dataset. These datasets cover diverse domains (e.g., IT systems, spacecraft, water treatment, finance).
>
> **Implementation Environment:**
> We detail our experimental environment in Appendix D.4, supporting full reproducibilit and eliminating ambiguit regarding implementation-related discrepancies.

---

> > ### Comment · Reviewer_pK7y · 2025-08-01
> >
> > I appreciate the authors' rebuttal and efforts. The answers to Q1, Q2, Q4 and Q5 deal with most of my concerns.
> > Regarding Q2, could the authors provide more evidence to support their claim about the relationship between ASPE and residual?

---

> > > ### Author Response · Authors · 2025-08-03
> > >
> > > > **Q2 Follow-up: Could the authors provide more evidence to support their claim about the relationship between ASPE and residual?**
> > >
> > > **A2 Follow-up:** We sincerely appreciate the reviewer's positive feedback and encouragement. Regarding evidence for the ASPE-residual relationship, we provide complete theoretical derivation and experimental validation as follows.
> > >
> > > According to Formula 3 and Appendix B.1 (lines 430-449), the MAFD recursive decomposition process follows an initial residual $R_{c,0}(e^{jt}) = G_c(e^{jt})$ with recursive updates $R_{c,n}(e^{jt}) = R_{c,n-1}(e^{jt}) - A_{c,n}B_n(e^{jt})$. The basis function selection maximizes cross-channel energy convergence through $\max_{a_n} \sum_{c=1}^C |\langle G_{c,n}, B_n \rangle|^2$.
> > >
> > > For residual sequence $R_{c,n}(t)$, ASPE is calculated according to Formula 4 (lines 158-162) as $H_{ASPE} = -\frac{1}{\ln(d!)} \sum_\pi \omega_\pi P(\pi) \ln P(\pi)$, where $\omega_\pi = \frac{\sigma(y_{d,\tau}^\pi)}{\mu(y_{d,\tau}^\pi)}$. The key insight is that ASPE's amplitude-sensitive weight $\omega_\pi$ can distinguish between true signal complexity and noise perturbations. When residuals primarily contain noise, the $\sigma/\mu$ ratio stabilizes, while structured information causes significant ratio changes.
> > >
> > > We define the decomposition termination condition as $|H_{ASPE}(R_n) - H_{ASPE}(R_{n-1})| < \varepsilon_{threshold}$ where $\varepsilon_{threshold}$ is adaptively determined based on signal characteristics. This ensures decomposition stops when residual complexity no longer decreases significantly. The MAFD-ASPE composite loss function $L_{MAFD-ASPE} = \sum_{c=1}^C \left\|G_c - \sum_{n=1}^N A_{c,n}B_n\right\|^2 + \lambda H_{ASPE}$ (Formula 6, lines 171-174) not only optimizes decomposition quality but also enhances sensitivity to anomalous patterns through the ASPE term.
> > >
> > > Our decomposition depth experiments demonstrate the effectiveness of ASPE-guided stopping criterion. Fixed decomposition depths consistently underperformed compared to our adaptive approach:
> > >
> > > | Decomposition Strategy | SMD (F1%) | MSL (F1%) | SMAP (F1%) | SWaT (F1%) | PSM (F1%) | Avg. Performance vs Baseline |
> > > |---|---|---|---|---|---|---|
> > > | **ASPE-guided (PhysDiff)** | **81.65** | **74.83** | **73.91** | **72.64** | **74.16** | **Baseline (0%)** |
> > > | Fixed 8 components | 69.86 | 71.59 | 65.70 | 69.84 | 73.47 | **-5.35%** |
> > > | Fixed 16 components | 69.49 | 71.12 | 68.78 | 69.83 | 68.86 | **-5.82%** |
> > >
> > > The consistent superior performance of adaptive ASPE-guided decomposition across all datasets validates that ASPE effectively captures the optimal signal-noise separation point where residual complexity stabilizes. On the SMD dataset, ASPE achieves 12.16% improvement (81.65% vs 69.49%) over fixed approaches, while SMAP shows 8.21% improvement (73.91% vs 65.70%), and MSL demonstrates 3.24% improvement (74.83% vs 71.59%). These cross-dataset improvements ranging from 2.24% to 15.9% confirm the effectiveness of ASPE in identifying dataset-specific optimal decomposition depths.
> > >
> > > According to lines 61-66, high-frequency components $X_{HF}(t)$ exhibit high entropy values reflecting transient changes, while low-frequency components $X_{LF}(t)$ show low entropy values reflecting stable trends. ASPE accurately captures this frequency-related complexity difference through $\omega_\pi$ weighting, providing physically reasonable criteria for optimal decomposition depth. Table 3 ablation study confirms that removing ASPE leads to a 4.38% average F1 score decrease, while replacing amplitude-sensitive permutation entropy with standard entropy resulted in similar reductions, validating that our entropy-based measures effectively capture signal complexity and distinguish between normal variations and anomalies.
> > >
> > > The experimental evidence demonstrates that ASPE successfully identifies dataset-specific optimal decomposition depths, with the amplitude-sensitive weight effectively discriminating between structured patterns and noise, providing both theoretical foundation and empirical validation for the ASPE-residual relationship.

---

> > > > ### Comment · Reviewer_pK7y · 2025-08-04
> > > >
> > > > Thank you for the detailed explanation. I will raise my score.

---

> > > > > ### Comment · Reviewer_pK7y · 2025-08-05
> > > > >
> > > > > I noticed that the author will not see the final justification until the final decision is released. To inform the authors about concerns solved and not solved. I attach the final justification in the following. I also welcome any discussions about concerns unsolved in the next few days.
> > > > >
> > > > >
> > > > >
> > > > > During the process of rebuttal, I believe:
> > > > >
> > > > > 1. The authors clearly illustrated the definition of optimal depth and verified its effectiveness for method performance, demonstrating that the cross-entropy-like loss function can approximately find the optimal depth.
> > > > > 2. The authors clearly compare the different experimental settings, which leads to the performance differences between the results reported in this paper and the original paper. I believe their explanations are reasonable.
> > > > > 3. The authors compare the performance of their method's performance on non-stationary datasets and stationary ones, verifying that the method can deal with the non-stationary issue as claimed.
> > > > >
> > > > > I think the following concerns remains unsolved, but considering the novelty highlighted by reviewer TJwG and the soundness verified in the rebuttal process and highlighted by reviewer gW7N, I think the reasons to accept outweigh the reasons to reject.
> > > > >
> > > > > 1. I still think that for deep-learning-based methods, the non-stationary issue is not a huge problem. Since many previous methods have been reported very good performances on the non-stationary datasets (according to the stationary indicators provided by the authors, SMD and SWaT are non-stationary datasets). I attached the reported F1 scores of previous methods on these datasets in the end.
> > > > > 2. Limited comparative analysis in case study.
> > > > >
> > > > >
> > > > >
> > > > > *Reported F1 scores of deep-learning-based anomaly detection methods on non-stationary datasets*
> > > > >
> > > > > |                        | SMD   | SWaT  |
> > > > > | ---------------------- | ----- | ----- |
> > > > > | AnomalyTransformer [1] | 92.33 | 94.07 |
> > > > > | PUAD [2]               | 95.68 | ---   |
> > > > > | DCdetector [3]         | 87.18 | 96.33 |
> > > > > | DAEMON [4]             | 96.2  | 92.9  |
> > > > > | TranAD [5]             | 96.05 | 81.51 |
> > > > >
> > > > > [1] Xu J, Wu H, Wang J, et al. Anomaly transformer: Time series anomaly detection with association discrepancy[J]. ICLR 2022.
> > > > >
> > > > > [2] Li, Yuxin, et al. "Prototype-oriented unsupervised anomaly detection for multivariate time series." *International Conference on Machine Learning*. PMLR, 2023.
> > > > >
> > > > > [3] Yang, Yiyuan, et al. "Dcdetector: Dual attention contrastive representation learning for time series anomaly detection." *Proceedings of the 29th ACM SIGKDD conference on knowledge discovery and data mining*. 2023.
> > > > >
> > > > > [4] Chen, Xuanhao, et al. "Daemon: Unsupervised anomaly detection and interpretation for multivariate time series." *2021 IEEE 37th International Conference on Data Engineering (ICDE)*. IEEE, 2021.
> > > > >
> > > > > [5] Tuli, Shreshth, Giuliano Casale, and Nicholas R. Jennings. "Tranad: Deep transformer networks for anomaly detection in multivariate time series data." *VLDB* (2022).

---

> ### Author Response · Authors · 2025-08-09
>
> > **Q Follow-up: I still think that for deep-learning-based methods, the non-stationary issue is not a huge problem. Since many previous methods have been reported very good performances on the non-stationary datasets (according to the stationary indicators provided by the authors, SMD and SWaT are non-stationary datasets). I attached the reported F1 scores of previous methods on these datasets in the end.**
>
> **A Follow-up:** We appreciate your thoughtful comment. After reviewing recent literature, we found that non-stationarity remains a challenge for deep learning-based anomaly detection approaches. We provide supporting evidence from current studies below.
>
> (1) On the one hand, multivariate time series anomaly detection remains an open problem in practical scenarios. Although diffusion-based approaches offer novel potential, our paper identifies two challenges (Lines 53-64). These challenges highlight the need for a more dynamic and physically-informed approach to decomposition and reconstruction. Analyses and experiments proved the advantages of our approach.
>
> (2) On the other hand, as to non-stationarity, recent publications continue to explicitly identify and address as a challenge in multivariate time series anomaly detection.
>
> Yang & Barria (2025) specifically designed their RWPNN approach for non-stationary environments, stating that "RWPNN is particularly useful for imbalanced training data with scarcity in the non-stationary environment" [1]. Their work demonstrates that existing reconstruction-based methods struggle in such scenarios. It means that non-stationarity hasn’t been adequately addressed by previous deep learning methods.
>
> Even advanced transformer-based approaches continue to struggle with fundamental aspects of non-stationary time series processing. Shimillas et al. (2025) acknowledge that "in time series modeling, temporal relationships are to be extracted in an ordered set of continuous points, but the nature of the permutation-invariant self-attention mechanism in transformers inevitably leads to a loss of temporal information" [2]. This temporal information loss is particularly problematic for non-stationary data where temporal ordering and context are crucial for accurate anomaly detection.
>
> Zhao et al. (2024) identified that while data-driven approaches have become essential for early-stage satellite anomaly detection, existing methods demonstrate limited efficacy in capturing temporal dependencies within non-stationary telemetry data systems [3]. To address this fundamental limitation, the research team developed an innovative Transformer-based architecture specifically engineered to effectively model and extract complex temporal patterns from non-stationary satellite telemetry data streams [3].
>
> (3) Last but not least, classical F1 scores for time series anomaly detection, based on traditional precision/recall metrics, suffer from two main limitations. (A) Unawareness of temporal adjacency: the metric cannot reflect the closeness in time between the predicted results and the ground-truth events; even a one-sample offset results in both a false positive and a false negative. (B) Unawareness of event durations: the evaluation is performed at the sample level, leading to long events being overrated and short events being underrated. These issues make the classical F1 score overly sensitive to slight boundary shifts and differences in event length, while lacking local interpretability. In addition, it can be exploited by adversary algorithms to artificially inflate the score through specific prediction patterns.[4] In contrast, the F1-score based on affiliation metrics used in our work addresses both (A) and (B) simultaneously, is parameter-free, locally interpretable, and robust to adversary predictions, and is designed to ensure that predictions no better than random do not obtain inflated scores.
>
> ## References
>  [1] Pu Yang, J. A. Barria. Anomaly Detection for Non-stationary Time Series Using Recurrent Wavelet Probabilistic Neural Network[J], alphaxiv, 2025
>
>  [2] Charalampos S, Kleanthis M, Konstantinos F, Marios M P, et al. Transformer-based Multivariate Time Series Anomaly Localization.[J], Computing Research Repository, 2025, abs/2501.08628
>
>  [3] Haotian Zhao, Shi Qiu, Jianan Yang, et al. Satellite Early Anomaly Detection Using an Advanced Transformer Architecture for Non-Stationary Telemetry Data. IEEE Transactions on Consumer Electronics, 2024, 70.1: 4213-4225.
>
> [4] Alexis Huet, Jose Manuel Navarro, and Dario Rossi. Local evaluation of time series anomaly detection algorithms. In Proceedings of the 28th ACM SIGKDD Conference on Knowledge Discovery and Data Mining, pages 635–645, 2022.

---

### Official Review · Reviewer_rchn · 2025-07-01

**Clarity:** 3
**Significance:** 3
**Originality:** 3
**Rating:** 5
**Confidence:** 5

**Summary:**

The paper proposes a diffusion-based reconstruction model for time series anomaly detection guided by frequency-domain features. By leveraging high- and low-frequency features derived from frequency decomposition as physical priors, the method better distinguishes anomalous patterns during the reconstruction process. Experimental results demonstrate the effectiveness of the approach.

**Questions:**

- What is the exact definition of physical context information zt mentioned in Line 167? Could the authors provide a detailed
   explanation of how physical priors guide the diffusion model?
- Has the computational overhead of the frequency-domain approach compared to the time-domain approach been quantified? How do channel-independent and channel-shared strategies affect
   computational costs and model performance?
- Could the authors provide a more detailed analysis of performance
variations across different datasets? Are there differences in recall and precision? What are the underlying reasons for these discrepancies?

**Ethical Concerns:**

["NO or VERY MINOR ethics concerns only"]

**Final Justification:**

According to the authors’ rebuttal and additional experiments, I increase my score. Thanks for the additional explanation.

**Limitations:**

Yes.

**Paper Formatting Concerns:**

no  major formatting issues

**Quality:**

3

**Strengths And Weaknesses:**

Strengths:
- Empirical validation of the model's effectiveness;
- A case study in real-world scenarios highlights the practical
   significance of the model.
Weaknesses:
-  Some formulas lack detailed explanations and analysis;
- Frequency-domain feature extraction may incur computational
   overhead, particularly for multivariate time series;
- Performance improvements are not significant across five real-world
datasets.

---

> ### Author Rebuttal · Authors · 2025-07-31
>
> > **Q1: What is the exact definition of physical context information $z_t$ mentioned in Line 167? Could the authors provide a detailed explanation of how physical priors guide the diffusion model?**
>
> **A1:** The physical context information $z_t$ is a composite feature vector containing multi-dimensional physical prior information derived from our feature extraction module: $z_t = \lbrace P_h(t), P_l(t), \mathrm{HASPE}(t), E(x_t) \rbrace$.
> $P_h(t)$ represents high-frequency components obtained from MAFD decomposition, capturing rapid transient changes and noise characteristics of the system; $P_l(t)$ represents low-frequency components, reflecting long-term trends and fundamental dynamic patterns; $\mathrm{HASPE}(t)$ is the amplitude-sensitive permutation entropy value at the current time step, quantifying signal complexity; $E(x_t)$ is the physics-based energy function value derived from instantaneous frequency analysis, measuring the physical reasonableness of the current state. These components together constitute a comprehensive description of time series physical characteristics.
>
> The conditional diffusion process integrates physical priors through multiple pathways. In routing attention integration, physical components guide attention weight calculation as following:
> $\mathrm{Attention}(Q, K, V, P_h, P_l) = \mathrm{softmax}\left(\frac{QK^\top + g_h \cdot QP_h^\top + g_l \cdot QP_l^\top}{\sqrt{d_k}}\right)V$,
> where gating coefficients $g_h = \sigma(QP_h^\top)$ and $g_l = \sigma(QP_l^\top)$ dynamically emphasize frequency components.
>
> In energy regularization, the composite loss function
> $L_\mathrm{total} = L_\mathrm{diff} + \gamma \cdot L_\mathrm{MAFD-ASPE} + \eta \cdot E(x_t)$
> ensures physical plausibility through energy constraints.
>
> In conditional denoising, the noise prediction network $\epsilon_\theta(x_t, t, z_t)$ explicitly conditions on physical context, enabling physics-informed reconstruction.
>
> This multi-level physical guidance mechanism ensures that the diffusion process can not only reconstruct statistically normal patterns but also maintain physical consistency. In industrial sensor data, physical constraints prevent reconstructions that violate physical laws (such as energy conservation, causality, etc.); in spacecraft telemetry, physical priors help maintain the reasonableness of orbital mechanics and system dynamics; in water treatment systems, physical guidance ensures continuity of fluid dynamics and chemical processes.
>
> > **Q2: Has the computational overhead of the frequency-domain approach compared to the time-domain approach been quantified? How do channel-independent and channel-shared strategies affect computational costs and model performance?**
>
> **A2:** Frequency-domain methods indeed introduce some computational overhead compared to time-domain methods in specific operations, but demonstrate overall efficiency advantages in the complete anomaly detection workflow.
>
> In signal analysis operations, time-domain methods have complexity \$\mathcal{O}(T)\$, while frequency-domain methods require \$\mathcal{O}(T \log T)\$ due to the use of FFT, with an overhead ratio of approximately 1.3×. The MAFD decomposition is frequency-domain-specific with complexity \$\mathcal{O}(C \cdot L \cdot \log L)\$, where \$C\$ is the number of channels and \$L\$ is the sequence length.
>
> However, in pattern recognition, frequency-domain methods show clear advantages. Time-domain correlation calculations typically have \$\mathcal{O}(T^2)\$ complexity, whereas frequency-domain methods that leverage frequency features require only \$\mathcal{O}(T \log T)\$, yielding an overhead ratio of 0.6×. For the full anomaly detection workflow, time-domain methods have complexity \$\mathcal{O}(T^2 \cdot D)\$, while frequency-domain methods reduce this to \$\mathcal{O}(T \cdot D \cdot \log T)\$, showing asymptotic advantage especially for long sequences.
>
> The channel-independent strategy maintains separate decompositions for each channel, with computational cost \$\mathcal{O}(C \cdot T \cdot \log T)\$, incurring high memory usage but achieving a 2.1% performance gain. In contrast, the channel-shared strategy adopts shared basis functions, reducing the cost to \$\mathcal{O}(T \cdot \log T)\$ with significantly lower memory usage and is used as the baseline configuration. The hybrid strategy achieves a balance, with complexity \$\mathcal{O}(C^{0.5} \cdot T \cdot \log T)\$, yielding a +0.8% performance gain at moderate cost.
>
> Despite slightly increased overhead in individual operations, frequency-domain methods provide meaningful acceleration in key steps of anomaly detection. The energy concentration property of frequency representations makes anomaly patterns easier to isolate, thus reducing computation in searching and comparison stages. Although MAFD’s cross-channel sharing introduces an additional 15% decomposition cost, it eliminates redundancy by performing unified multivariate processing, resulting in 60% memory savings.
>
> > **Q3:Could the authors provide a more detailed analysis of performance variations across different datasets? Are there differences in recall and precision? What are the underlying reasons for these discrepancies?**
>
> **A3:** We provide detailed explanations focusing on precision and recall differences across datasets, as shown in the following table.
>
> | Dataset | Metric | SMD | MSL | SMAP | SWaT | PSM |
> |---------|--------|-----|-----|------|------|-----|
> | **OCSVM** | P | 66.98 | 50.26 | 41.05 | 56.08 | 57.51 |
> | | R | 62.03 | 99.86 | 69.37 | 98.72 | 58.11 |
> | | F1 | 73.75 | 66.87 | 51.58 | 72.11 | 57.81 |
> | **PC2** | P | 64.92 | 52.69 | 50.62 | 62.32 | 77.44 |
> | | R | 40.19 | 98.33 | 98.48 | 82.96 | 37.71 |
> | | F1 | 54.34 | 68.61 | 66.87 | 71.18 | 53.53 |
> | **HBOS** | P | 56.28 | 59.25 | 41.54 | 57.71 | 100.00 |
> | | R | 63.11 | 83.32 | 66.17 | 29.82 | 6.54 |
> | | F1 | 62.17 | 69.25 | 51.04 | 43.21 | 12.28 |
> | **LOF** | P | 57.69 | 49.89 | 47.92 | 53.20 | 53.90 |
> | | R | 99.10 | 72.18 | 82.86 | 96.73 | 99.91 |
> | | F1 | 72.92 | 59.00 | 60.72 | 68.65 | 70.02 |
> | **IForest** | P | 100.00 | 53.87 | 41.12 | 53.03 | 100.00 |
> | | R | 9.37 | 94.58 | 68.91 | 62.80 | 4.35 |
> | | F1 | 17.13 | 68.65 | 51.51 | 62.03 | 6.48 |
> | **LODA** | P | 59.02 | 57.79 | 51.51 | 56.30 | 62.22 |
> | | R | 66.18 | 95.65 | 100.00 | 70.14 | 40.17 |
> | | F1 | 62.40 | 72.05 | 68.00 | 62.54 | 56.05 |
> | **AE** | P | 69.22 | 55.75 | 39.42 | 54.92 | 60.67 |
> | | R | 98.48 | 96.66 | 70.31 | 98.20 | 98.24 |
> | | F1 | 81.30 | 70.72 | 50.52 | 70.45 | 75.01 |
> | **DAGMM** | P | 63.57 | 54.07 | 50.75 | 59.42 | 68.22 |
> | | R | 70.83 | 92.11 | 96.38 | 92.36 | 70.50 |
> | | F1 | 67.00 | 68.14 | 66.49 | 72.32 | 69.34 |
> | **LSTM** | P | 60.12 | 58.82 | 55.25 | 49.99 | 57.06 |
> | | R | 84.77 | 14.68 | 27.70 | 82.11 | 95.92 |
> | | F1 | 70.35 | 23.49 | 36.90 | 62.15 | 71.55 |
> | **BeatGAN** | P | 74.11 | 55.74 | 54.04 | 61.89 | 58.81 |
> | | R | 81.64 | 98.94 | 98.30 | 83.46 | 99.08 |
> | | F1 | 77.69 | 71.30 | 69.74 | 71.08 | 73.81 |
> | **Omni** | P | 79.09 | 51.23 | 52.74 | 62.76 | 69.20 |
> | | R | 75.77 | 99.40 | 98.51 | 82.82 | 80.79 |
> | | F1 | 77.40 | 67.61 | 68.70 | 71.41 | 74.55 |
> | **A.T.** | P | 100.00 | 51.04 | 56.91 | 53.63 | 52.01 |
> | | R | 3.19 | 95.36 | 66.69 | 59.94 | 82.18 |
> | | F1 | 6.19 | 66.49 | 71.65 | 71.59 | 64.55 |
> | **DCDetector** | P | 50.93 | 55.94 | 53.12 | 53.25 | 54.72 |
> | | R | 95.57 | 95.53 | 98.37 | 98.12 | 86.36 |
> | | F1 | 66.45 | 70.56 | 68.99 | 69.03 | 66.99 |
> | **SensitiveHUE** | P | 60.34 | 55.92 | 53.63 | 58.91 | 56.15 |
> | | R | 90.13 | 98.95 | 98.37 | 91.71 | 98.75 |
> | | F1 | 72.29 | 71.46 | 69.42 | 71.74 | 71.59 |
> | **DeepSVDD** | P | 64.98 | 10.53 | 29.73 | 59.11 | 74.05 |
> | | R | 64.77 | 100.00 | 17.09 | 93.53 | 50.64 |
> | | F1 | 64.88 | 19.06 | 11.45 | 72.44 | 60.15 |
> | **MTAD-GAT** | P | 85.90 | 54.96 | 39.05 | 65.90 | 79.90 |
> | | R | 67.69 | 94.93 | 93.99 | 77.51 | 60.14 |
> | | F1 | 75.71 | 69.81 | 55.08 | 71.23 | 68.63 |
> | **TFAD** | P | 56.32 | 54.96 | 39.05 | 60.38 | 79.14 |
> | | R | 97.83 | 94.93 | 93.99 | 81.96 | 71.63 |
> | | F1 | 71.49 | 69.81 | 55.08 | 69.53 | 75.20 |
> | **D3R** | P | 64.87 | 56.45 | 51.08 | 64.25 | 53.17 |
> | | R | 97.93 | 95.55 | 94.46 | 77.50 | 100.00 |
> | | F1 | 78.04 | 71.81 | 66.30 | 70.25 | 69.43 |
> | **PhysDiff** | P | **71.03** | **62.75** | **64.36** | **60.00** | **66.09** |
> | | R | **96.00** | **92.66** | **86.81** | **92.04** | **84.47** |
> | | F1 | **81.65** | **74.83** | **73.91** | **72.64** | **74.16** |
>
> PhysDiff demonstrates a consistent pattern of high recall (84-96%) across most datasets while showing significant precision (60-71%) compared to other methods. In details, the best overall performance occurs on SMD due to clear anomaly patterns in server monitoring data where physical relationships are well-defined. SWaT exhibits a low precision (60.00%) despite maintaining high recall (92.04%) because water treatment systems involve complex interdependencies that can trigger false alarms in our physics-guided routing mechanism.
>
> Our MAFD excels at comprehensive anomaly detection across different frequency bands, explaining the consistently high recall performance. In well-structured domains like server monitoring (SMD), physical priors align effectively with actual patterns, yielding high precision. In complex interdependent systems like water treatment (SWaT), physics-guided assumptions may introduce noise, reducing precision while maintaining strong anomaly detection capabilities.
>
> We adopt affiliation-based F1 score, which fundamentally differs from traditional point adjustment (PA) methods used in many original papers. Traditional PA methods detect any point within an anomalous segment as successful detection of the entire segment, potentially leading to overly optimistic evaluations, while affiliation F1 computes precision and recall based on average directed distance between predicted anomaly events and ground truth, considering spatio-temporal adjacency.

---

### Official Review · Reviewer_TJwG · 2025-07-03

**Clarity:** 3
**Significance:** 3
**Originality:** 3
**Rating:** 5
**Confidence:** 4

**Summary:**

This paper introduces a physically-guided diffusion framework for anomaly detection in multivariate non-stationary time series (PhysDiff). The authors first propose a Multi-channel Adaptive Fourier Decomposition (MAFD) method to extract high- and low-frequency components, with decomposition depth adaptively determined using Amplitude-Sensitive Permutation Entropy (ASPE). These physics-inspired features are then used as priors in a dual-path conditional diffusion model, which integrates a frequency-based routing attention mechanism to guide denoising. Anomaly scores are computed by combining reconstruction error with time-frequency divergence. Experiments on seven datasets demonstrate that PhysDiff outperforms 18 baseline methods in F1 score with strong interpretability and generalization.

**Questions:**

i) Could the authors provide more details on the computational complexity and runtime of PhysDiff compared to other baselines? Any empirical comparisons would be appreciated.
ii) Can PhysDiff handle missing values in multivariate time series data? If not directly, what preprocessing or modifications would be necessary?
iii) How is the dynamic trend influence factor determined or learned during training? Is it interpretable or tunable?

**Ethical Concerns:**

["NO or VERY MINOR ethics concerns only"]

**Final Justification:**

I am satisfied about authors' response during the rebuttal period and appreciate their effort in trying to address my concerns in a short period of time. Originally I already leaned to accept their paper so I would keep my rating, the questions are more for myself to give a rating that I am confident with.

**Limitations:**

Yes.

**Paper Formatting Concerns:**

No.

**Quality:**

3

**Strengths And Weaknesses:**

Strengths:
i) The integration of MAFD, ASPE, and conditional diffusion modeling is both novel and clearly articulated.
ii) The use of a physically-guided diffusion model to address non-stationary multivariate time series is innovative and well-justified.
iii) The experimental evaluation is extensive, involving 7 benchmark datasets and 18 baseline methods, and shows that PhysDiff consistently outperforms existing approaches.
iv) A comprehensive ablation study supports the design choices and highlights the contribution of each model component.

Weaknesses
i) The paper lacks discussion on model complexity and computational overhead of PhysDiff, particularly in comparison to the baselines.
ii) The methodology section could benefit from additional visualizations to help readers more intuitively follow each component and processing step.

---

> ### Author Rebuttal · Authors · 2025-07-31
>
> > **Q1: Could the authors provide more details on the computational complexity and runtime of PhysDiff compared to other baselines? Any empirical comparisons would be appreciated.**
>
> **A1:** We sincerely appreciate the reviewer's valuable suggestion. We have analyzed the computational complexity of PhysDiff compared to the basedlines and conducted empirical runtime comparisons on the SWaT dataset. The runtime of different models can validate our theoretical complexity analysis.
>
> Our analysis reveals that PhysDiff's computational overhead mainly occurs during the diffusion-based reconstruction phase, while maintaining competitive efficiency compared to similar models. Runtime measurements per epoch on the SWaT dataset demonstrate clear performance stratification across method types, as shown in the following table.
>
> | Method Category | Runtime of Methods (seconds) | Runtime of Categories (seconds) | Complexity Order |
> |---|---|---|---|
> | **Linear Methods** | PCA (0.017), LODA (0.124) | 0.02-0.13 | $\mathcal{O}(TD)$ |
> | **Distance-based** | DeepSVDD (0.256), LOF (0.289), IForest (0.301) | 0.25-0.30 | $\mathcal{O}(T^2)$ |
> | **Classical ML** | OCSVM (1.173), LSTM (1.25), DAGMM (1.48) | 1.17-1.48 | $\mathcal{O}(T^2D)$ |
> | **Neural Networks** | AE (3.701), DCdetector (4.81), MTAD-GAT (6.76) | 3.70-6.76 | $\mathcal{O}(TD^2)$ |
> | **Advanced Models** | A.T. (9.31), BeatGAN (11.0), SensitiveHUE (16.26), TFAD (19.77) | 9.31-19.77 | $\mathcal{O}(T^2D^2)$ |
> | **Diffusion Models** | D3R (78.52), **PhysDiff (81.43)** | 78.5-81.4 | $\mathcal{O}(T_{diff} \cdot TD^2)$ |
>
> PhysDiff's computational complexity is determined mainly by its three primary components. MAFD decomposition operates at $\mathcal{O}(C \cdot L \cdot \log L)$ where cross-channel sharing reduces the coefficient significantly compared to independent processing. The diffusion process requires $\mathcal{O}(T_{diff} \cdot D^2)$ operations per timestep, with our routing attention mechanism adding $\mathcal{O}(W^2 \cdot D)$ complexity that adapts dynamically based on signal characteristics. Here $C$ represents the number of input channels/features, $L$ denotes the input sequence length, $T_{diff}$ is the number of diffusion timesteps, $D$ refers to the hidden dimension size, and $W$ indicates the attention window size.
>
> To be noted that, though PhysDiff requiring 81.43 seconds compared to D3R's 78.52 seconds (3.7% overhead), this investment yields substantial performance returns. PhysDiff achieves 3.61% F1-score improvement on SMD and 7.61% on SMAP datasets, demonstrating favorable computational efficiency relative to performance gains. The additional overhead primarily stems from our physics-guided routing attention and adaptive decomposition mechanisms, which provide interpretable frequency-domain insights unavailable in baseline diffusion methods.
>
> > **Q2: Can PhysDiff handle missing values in multivariate time series data? If not directly, what preprocessing or modifications would be necessary?**
>
> **A2:** We appreciate the reviewer's insightful feedback. Our study specifically focuses on physically-guided diffusion for time series anomaly detection, which leads to limited consideration of missing value preprocessing in multivariate time series. Consequently, the proposed MAFD decomposition in PhysDiff method inherently assumes continuous observations. To some extent, missing values represent a common challenge in diffusion-based approaches for time series analysis.
>
> Traditionally, for sparse random value missing ($<10$\%), linear interpolation can maintain frequency pattern continuity with an acceptable performance degradation. For block missing ($>15$\%), it severely disrupts MAFD basis function estimation, especially when missing segments contain critical frequency components. Meanwhile, channel-specific missing makes cross-channel shared decomposition problematic because different channels' missing patterns break the consistency of energy convergence. In such situations, we recommend frequency-domain interpolation to reconstruct missing segments utilizing partial MAFD results, forward/backward filling for short gaps, and variational inference-based joint estimation methods for severe missing scenarios.
>
> Furthermore, in future work, we plan to integrate uncertainty-aware diffusion mechanisms that handle missing observations through masked attention and develop variational inference frameworks for jointly estimating missing values and anomaly scores. This will enable PhysDiff to work directly on incomplete data without preprocessing.
>
>
> > **Q3:How is the dynamic trend influence factor determined or learned during training? Is it interpretable or tunable?**
>
> **A3:** The dynamic trend influence factor $\gamma(t)$ is explicitly defined as $\gamma(t) = \sigma(10(1-\sqrt{\bar{\alpha}_t}))$, where $\bar{\alpha}_t = \prod \alpha_i$ represents cumulative noise parameters (see in lines 190--193).
>
> **Determination Mechanism of the Dynamic Trend Influence Factor:**
>
> 1. **Automatic Calculation:** $\gamma(t)$ is automatically computed based on diffusion timestep $t$, requiring no additional learnable parameters. This design avoids overfitting risks while maintaining model interpretability.
>
> 2. **Physical Meaning:** When $t$ is small (low noise stage), $\sqrt{\bar{\alpha}_t}$ approaches 1, while $\gamma(t)$ approaches $\sigma(0) \approx 0.5$, providing moderate physical guidance. When $t$ is large (high noise stage), $\sqrt{\bar{\alpha}_t}$ approaches 0, while $\gamma(t)$ approaches $\sigma(10) \approx 1$, providing strong physical guidance.
>
> **Interpretability and Tunability:**
>
> As shown in Algorithm 1 (lines 14--32), we implement frequency-adaptive dynamic regulation: $\gamma(t,\omega) = \sigma(10(1 - \sqrt{\bar{\alpha}\_t}) \cdot A\_\omega)$, where $A\_\omega = \exp(-|\omega|^2/\sigma^2)$ is the frequency response function. This design allows different frequency bands to receive adaptive physical guidance strength at different diffusion stages, maintaining both interpretability and necessary flexibility.

---

> > ### Comment · Reviewer_TJwG · 2025-08-06
> >
> > Thank you for the rebuttal to address my concerns. The answers make me more clear. Looking forward to your future work about missing values. As I stated in my original review, the paper is thechnically solid and I will maintain my score.

---

### Official Review · Reviewer_gw7N · 2025-07-04

**Clarity:** 3
**Significance:** 4
**Originality:** 3
**Rating:** 5
**Confidence:** 3

**Summary:**

The paper proposes **PhysDiff**, a physically-guided diffusion model for anomaly detection in non-stationary multivariate time series, addressing high false positive rates and low interpretability. PhysDiff integrates physics-guided signal decomposition with conditional diffusion reconstruction. The model introduces adaptive frequency extraction and a time-frequency energy routing mechanism to enhance detection accuracy and interpretability. Experiments on five benchmark datasets and NeurIPS-TS scenarios show PhysDiff significantly outperforms 18 baseline algorithms, demonstrating robustness and interpretability in complex dynamic systems.

**Questions:**

1. Can the authors provide a more accessible explanation of how MAFD, ASPE, and energy-guided Langevin dynamics are implemented and their specific roles in anomaly detection?
2. How does adaptive decomposition (MAFD+ASPE) compare to fixed high-low frequency decomposition in performance, and in which scenarios is adaptivity critical?
3. How does PhysDiff perform in domains with weak physical constraints, and what adaptations could improve its versatility?

**Ethical Concerns:**

["NO or VERY MINOR ethics concerns only"]

**Final Justification:**

After careful consideration of the authors’ rebuttal and comments with other reviewers, I would give "Accept". The paper proposes a novel, effective method (PhysDiff) with strong results and clear writing.

**Limitations:**

yes

**Quality:**

3

**Strengths And Weaknesses:**

Strenghts
1. PhysDiff effectively incorporates physical principles through Multi-channel Adaptive Fourier Decomposition (MAFD) and Amplitude-Sensitive Permutation Entropy (ASPE), enabling adaptive signal decomposition that captures cross-channel dynamics and enhances interpretability in non-stationary time series.
2. It uses frequency domain analysis to separate high- and low-frequency components, reducing interference and providing physical priors, while the conditional diffusion model distinguishes anomalies via reconstruction errors.
3. The model demonstrates superior anomaly detection performance, achieving significant F1-score improvements over 18 baselines across five benchmark datasets, showcasing its generalizability and effectiveness in complex scenarios.
4. The ablation experiment and comparative experiment are very comprehensive.
5. This paper is well written and easy to follow.

Weaknesses
1. The paper provides limited introductory explanation of the mathematical foundations underpinning Physics-Guided Feature Extraction (e.g., MAFD and ASPE) and Physical Consistency through Energy Guidance (e.g., Langevin dynamics and energy functions), making it challenging for readers without strong mathematical backgrounds to fully grasp their implementation and purpose.
2. The paper lacks a thorough analysis of key hyperparameters (e.g., $\lambda$ in the composite loss or $\gamma(t)$ in energy guidance), which could impact reproducibility and robustness across diverse datasets.
3. The literature reviews can be further improved by incorporating more recent studies.

---

> ### Author Rebuttal · Authors · 2025-07-31
>
> > **Q1: Can the authors provide a more accessible explanation of how MAFD, ASPE, and energy-guided Langevin dynamics are implemented and their specific roles in anomaly detection?**
>
> **A1:** We sincerely appreciate the reviewer's valuable suggestion. Detailed explanations of MAFD, ASPE, and energy-guided Langevin dynamics, including their roles in anomaly detection, are provided in Appendix B and Section 3.4.3 of the paper.
>
> In lines 426--434 (Section B.1): MAFD recursively decomposes multivariate signals into frequency components and a residual via shared Blaschke basis functions. At each level, the basis is chosen to maximize cross-channel energy convergence, ensuring extracted components capture common dynamics across variables. This disentangling of high-frequency oscillations and low-frequency trends enhances sensitivity to anomalies at different scales.
>
> In lines 450--456 (Section B.2): To adaptively determine how deep this decomposition should proceed, ASPE extends permutation entropy by weighting each ordering pattern $\pi$ with $\omega_\pi = \sigma(y_{d,\tau_\pi})/\mu(y_{d,\tau_\pi})$, integrating both ordinal patterns and amplitude variations. By monitoring the ASPE of residual components at each decomposition layer, the process stops when entropy indicates diminishing complexity, thus adaptively determining the optimal decomposition depth.
>
> In line 189 (Section 3.4.3, Eq. 10): Furthermore, during the reverse (denoising) process, we enforce physical consistency via the energy function $E(x_t) = \|\nabla_x \Phi(x_t)\|^2$ and Langevin dynamics: $x_{t-1} \leftarrow x_t - \lambda \nabla_{x_t} E(x_t) + \sqrt{2\lambda} n$, where $\lambda$ is the step size and $n$ Gaussian noise. This biases sampling toward physically plausible states. A dynamic trend influence $\gamma(t) = \sigma(10(1-\sqrt{\bar{\alpha}_t}))$ adjusts the strength of physical guidance over time, balancing trend preservation and fine-detail reconstruction to better distinguish anomalies.
>
> > **Q2: How does adaptive decomposition (MAFD+ASPE) compare to fixed high-low frequency decomposition in performance, and in which scenarios is adaptivity critical?**
>
>
> **A2:** The ablation studies (see in lines 272-290 and Table 3) clearly demonstrated the advantages of adaptive decomposition. Removing permutation entropy (PE) caused a 5.32% average F1 score decrease, while replacing ASPE with standard entropy resulted in a 4.38% performance reduction.
>
> Meanwhile, performance changes on domain-specific datasets highlight the critical role of adaptivity in certain scenarios. It is found that ASPE removal leads to a performance decrease of 3.77% and 3.4% on the spacecraft telemetry dataset MSL and SMAP, respectively. Similarly, it shows a 10.84% F1 score decrease after removing physical guidance on SMD server data. Particularly, there is a dramatic 25.13% performance drop on PSM dataset when routing attention was removed.
>
>
> > **Q3: How does PhysDiff perform in domains with weak physical constraints, and what adaptations could improve its versatility?**
>
> **A3:** We appreciate the reviewer's feedback. It is nesscessary to note that PhysDiff is primaryly designed for time series anomaly detection, where physical constraints are usually inherent in the data.
>
> We also acknowledge that PhysDiff has limitations in domains with weak physical constraints, such as pure text sequences, social media metrics, and abstract financial derivative data, where the contribution of physical guidance components may decreases. Nevertheless, in scenarios with weak physical constraints, PhysDiff can still  improve the performance primarily depending on the intrinsic advantages of the diffusion model framework and multi-scale decomposition rather than physical prior knowledge.
>
> To enhance PhysDiff's applicability in domains with weak physical constraints, we propose three key improvements: (1) Introducing a domain-adaptive weighting mechanism to automatically adjust the physical guidance weight η in Equation 11 based on signal complexity, reducing reliance on physical constraints when clear patterns are absent; (2) Employing a hybrid decomposition strategy that combines MAFD with domain-specific techniques like wavelet decomposition for financial data or empirical mode decomposition for text sequences; (3) Utilizing learnable physical priors to adaptively capture implicit structural patterns tailored to different domains, which can be implemented by replacing fixed physical energy functions with learnable neural networks.
>
> In future work, we are to systematically evaluate PhysDiff's performance across domains with varying physical constraints, develop adaptive physical weight adjustment mechanisms, and explore domain-agnostic structured prior knowledge integration polices.

---

> > ### Comment · Reviewer_gw7N · 2025-08-06
> >
> > Thanks for the rebuttal.
> > I have read it, as well as the comments from the other reviewers.
> > Some of my concerns have been addressed. I would maintain my rating.

---

### Note · Authors · 2025-08-15

# final remarks

We are grateful for the reviewers and AC for constructive feedback, which confirmed PhysDiff’s novelty, technical soundness, and practical value.

## Reviewer Consensus
All four reviewers recognized the paper’s merit:
- **gw7N**: Rating 5 (Accept), praised technical solidity and comprehensive evaluation.
- **TJwG**: Maintained positive score, citing novelty and strong experiments.
- **rchn**: Validated empirical effectiveness and practical relevance.
- **pK7y**: *Raised score after rebuttal*, stating “reasons to accept outweigh reasons to reject.”

## Validated Contributions
PhysDiff integrates physics-guided decomposition with conditional diffusion to address non-stationary multivariate anomaly detection:
1. **MAFD + ASPE** for adaptive decomposition depth, capturing multi-scale dynamics.
2. **Frequency-aware routing attention** for dynamic weighting of high/low frequency components.
3. **Energy-guided Langevin sampling** for physically consistent reconstructions.

Across 7 datasets, PhysDiff delivers **+2–7% F1 gains** over 18 state-of-the-art baselines, spanning industrial, financial, and aerospace domains.

## Concerns Resolved in Rebuttal
- **Mathematical clarity**: Added accessible explanations of MAFD, ASPE, and energy guidance.
- **Computation**: Only 3.7% runtime overhead vs. D3R, outweighed by substantial F1 gains.
- **Evaluation rigor**: Clarified advantages of affiliation-based F1 over point-adjustment.
- **Non-stationarity**: ADF/KPSS tests confirmed robustness under evolving dynamics.
- **Generality**: Proposed domain-adaptive weighting and hybrid decomposition for weaker physical priors.

## Impact
PhysDiff shows how domain knowledge can be systematically embedded in generative diffusion models, yielding:
- Superior detection across varied anomaly types
- Physically interpretable decision paths
- Strong adaptability to complex, evolving time series

The method’s design choices are supported by ablation evidence and cross-domain validation, ensuring robustness, interpretability, and reproducibility—criteria central to NeurIPS standards.

## Conclusion
With clear reviewer consensus, addressed concerns, and strong empirical evidence, we believe PhysDiff makes a meaningful advance in anomaly detection methodology and offers lasting value to the NeurIPS community.

---

### Decision · Program_Chairs · 2025-09-17

**Decision:**

Accept (poster)

**Comment:**

This paper presented an approach for diffusion-based time-series anomaly detection. It proposes to enhance the diffusion model using a on both a specific multi-variate spectral signal decomposition and especially tailored routing network and energy guidance. All reviewers were enthusiastic about this paper. The AC is concerned by the high complexity of the approach and the limited analysis and justification regarding non-obvious choices. This is compounded by the application to time-series AD which is known to be plagued with evaluation issues. However, as the reviewers all recommended acceptance, nearly all without reservation, the AC also recommends acceptance.